https://doi.org/10.1038/s41467-019-13741-x · **OPEN**

# Microbial Fe(III) reduction as a potential iron source from Holocene sediments beneath Larsen Ice Shelf

Jaewoo Jung [1], Kyu-Cheul Yoo[2], Brad E. Rosenheim[3], Tim M. Conway [3,4], Jae Il Lee[2], Ho Il Yoon[2], Chung Yeon Hwang [2], Kiho Yang [1], Christina Subt[3] & Jinwook Kim[1]*

Recent recession of the Larsen Ice Shelf C has revealed microbial alterations of illite in marine sediments, a process typically thought to occur during low-grade metamorphism. *In situ* breakdown of illite provides a previously-unobserved pathway for the release of dissolved $Fe^{2+}$ to porewaters, thus enhancing clay-rich Antarctic sub-ice shelf sediments as an important source of Fe to Fe-limited surface Southern Ocean waters during ice shelf retreat after the Last Glacial Maximum. When sediments are underneath the ice shelf, $Fe^{2+}$ from microbial reductive dissolution of illite/Fe-oxides may be exported to the water column. However, the initiation of an oxygenated, bioturbated sediment under receding ice shelves may oxidize Fe within surface porewaters, decreasing dissolved $Fe^{2+}$ export to the ocean. Thus, we identify another ice-sheet feedback intimately tied to iron biogeochemistry during climate transitions. Further constraints on the geographical extent of this process will impact our understanding of iron-carbon feedbacks during major deglaciations.

[1] Department of Earth System Sciences, Yonsei University, Seoul 03722, Korea. [2] Korea Polar Research Institute, Incheon 21990, Korea. [3] College of Marine Science, University of South Florida, Tampa, FL, USA. [4] School of Geosciences, University of South Florida, Tampa, FL, USA. *email: jinwook@yonsei.ac.kr

The limited availability of iron (Fe) in the surface Southern Ocean, compared to major nutrients such as nitrate and phosphate, leads to underutilization and thus outgassing of upwelled $CO_2$ in some areas of the modern Southern Ocean[1,2]. Relief of this Fe limitation, and sequestration of carbon into the deep ocean, is thought to enhance the Southern Ocean's ability to act as a mediator of global climate over millennial to ice-age timescales[3,4]. However, the source, transport, and fate of Fe in the Southern Ocean has been widely debated, with sources ranging from dust, ice sheets, iceberg-rafted debris (IRD) to sub-ice shelf, and other continental shelf sediments[5]. With evidence supporting spatially variable contributions of Fe from numerous sources to the Southern Ocean, consideration of the configuration of ice shelves and their dynamics is vital for understanding how changes in Fe supply help to drive carbon uptake in the Southern Ocean.

The Antarctic Peninsula, one of the fastest-warming regions on the Earth[6], acts as a sentinel for changes in Antarctic ice dynamics. Since the short period of satellite observations, rapid disintegration of ice shelves and retreat of continental glaciers has been observed on the Antarctic Peninsula and linked to potential increases in sea level rise[7]. Larsen ice shelves A and B (LIS-A and LIS-B) disintegrated catastrophically in 1995 and 2002, respectively, whereas the largest of the peninsular ice shelves, LIS-C, has persisted, decreasing in the size, and thinning through the Holocene to the present[8] (Fig. 1). Similar to many other places in Antarctica[9], grounding line retreat for LIS-C has been observed to be faster than the average rate since the Last Glacial Maximum[9] (LGM) of 25 m/year, particularly between 2010 and 2016.

Environmental change on the Antarctic Peninsula also significantly impacts the redox conditions of surface sediments during the retreat of ice shelves, driven by the overlying bottom waters changing from relatively stagnant to open-ocean oxygenated conditions[10,11]. These redox changes in the sediments may cause a variation in clay mineral alteration[12,13]. Around 50% of the clay minerals of Antarctic sub-ice shelf sediments are composed of illite[14], which contains redox-sensitive Fe in its crystal structure, and is thus potentially sensitive to changes in sediment redox conditions. Traditionally, however, alteration of illite crystallinity (IC) has been assumed to require temperature and pressure, and thus be restricted to low-grade metamorphic settings[15,16]. Recently, although, in situ alteration of illite has been identified beyond such metamorphic settings, and used to reconstruct paleoclimate conditions such as the Holocene monsoonal weathering, for example, Yangtze River[17], and Southwest Indian continental shelf[18]. In these studies, chemical weathering in organic-rich sediments is thought to drive the IC alteration, accelerated by wet and warm monsoonal conditions. However, none of these studies have considered the possibility of microbial alteration of illite at low temperatures. Microbially induced geochemical reactions can modify both mineral structure and chemistry[13], properties that are sensitive to redox conditions[12], and microbes are present in cold but diverse glacio-marine environments[19] associated with ice sheets[20], introducing the possibility that microbes may play a role in biogeochemical weathering and mineral alteration, even at very low temperatures[21]. Whereas a range of iron minerals are known to be sources of dissolved Fe upon breakdown by iron-reducing or -oxidizing bacteria[22,23], adding illite to this group, and at low temperatures, provides a potential new broadly distributed mechanism of dissolved $Fe^{2+}$ to sediment porewaters from the microbial alteration of clay minerals[24,25]. This process is distinct from reductive Fe production by microbial respiration of Fe hydroxides[26].

Here we demonstrate that microbe–illite interactions can be significant at low temperatures and pressures, affecting dissolved $Fe^{2+}$ release to porewaters and the overlying water column and thus productivity and carbon cycling in the Southern Ocean. As such, we present a broadly applicable alternative pathway for dissolved $Fe^{2+}$ production in sediment porewaters through microbial alteration of clay minerals, in addition to Fe(III) oxides, the strength of which will change corresponding to the depositional conditions during the expansion and retreat of ice shelves.

## Results and discussion

**Lithological setting.** To address the possibility of IC changes sourcing Fe to the water column beneath an ice shelf, we collected a 2.38-m-long marine sediment core (EAP13-GC16B) on the northwestern part of the LIS-C embayment (Fig. 1) aboard the icebreaker R/V *Araon* in 2013 (ANA03C Cruise). This core from the continental shelf provides us with information on changes in sedimentary environments during deglaciation, such as the proximity and stability of ice shelves and the influx of meltwater and terrigenous sediments[27]. The sedimentary sequence unveiled adjacent to LIS-C preserved a depositional record through the Holocene at shallow core depths (<3 m), below the temperatures and pressures of metamorphic modifications[28]. As such, our core represents an excellent test for the capability of biogeochemical reactions to alter the mineralogical characteristics of the subsurface Antarctic shelf sediments during the Holocene. Core EAP13-GC16B is defined by four distinct lithological units (U1–4) (Fig. 2a), with the upper unit U1 being characterized by sandy clay and IRD, foraminifera and diatoms, U2 and U3 by well-laminated silty clays, and U4 by sandy diamicton with cobbles, with a sharp boundary between U3 and U4 (Fig. 2b). These sedimentary sequences span the transitions from glacial through sub-ice shelf ($6000 ± 420$, $8000 ± 410$, and $11,500 ± 470$ cal. years BP at 85, 95, and 192 cm; U2–3) to occasionally open marine conditions ($1660 ± 70$, and $4140 ± 70$ cal. years BP at 0 and 14 cm; U1) since the LGM (Fig. 2c–e), based on calibrated dates derived from application of Ramped PyrOx $^{14}C$[29]. The dark gray sandy diamicton present in U4 is interpreted as being rapidly deposited

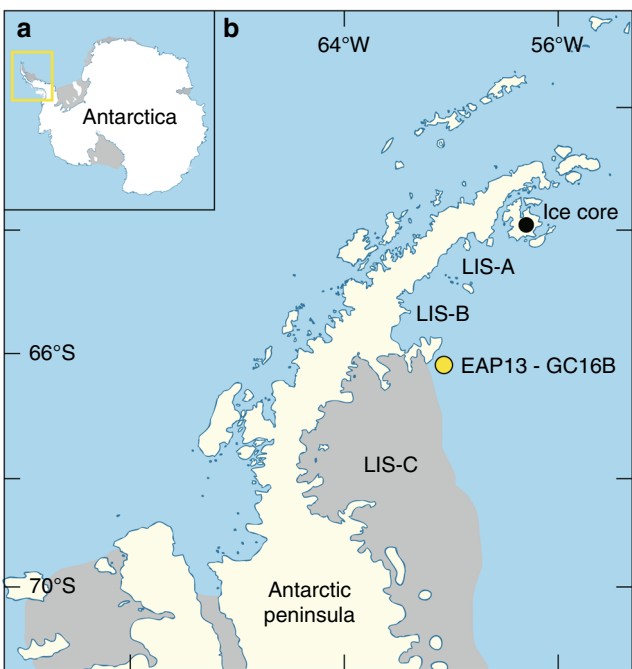

**Fig. 1 Location map of study area in Antarctica. a** Study area (yellow box) along the northern Antarctic Peninsula. **b** EAP13-GC16B sediment core collected from Larsen ice shelf C (LIS-C) embayment. Ice core drilling site[7], James Ross Island is north of Larsen ice shelves A (LIS-A) and B (LIS-B).

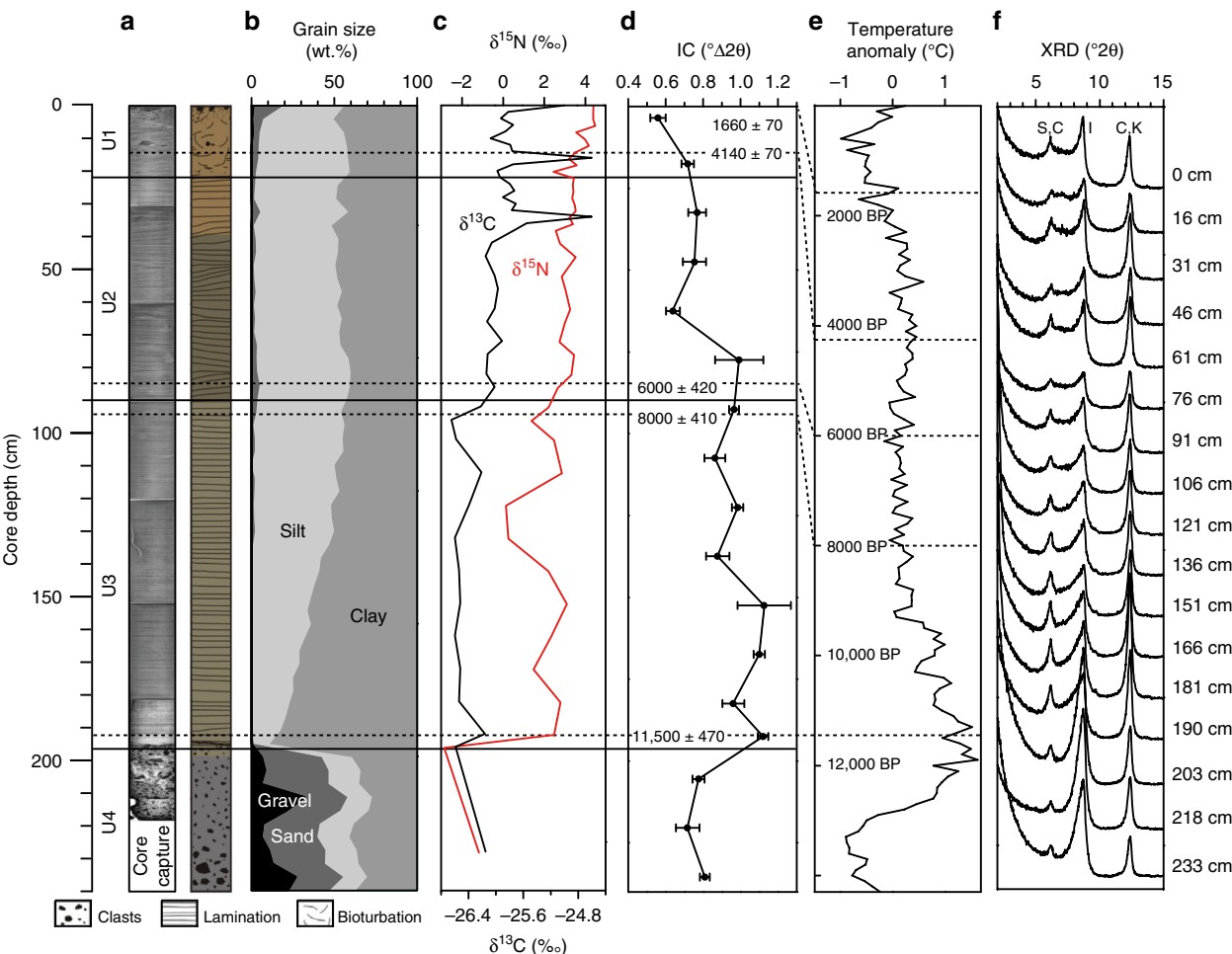

**Fig. 2 Downcore variations of EAP13-GC16B. a** Sedimentary facies (four distinct lithological units from top (U1) to bottom (U4)) with X-ray core image, **b** grain-size proportion, **c** isotopic composition ($\delta^{13}C$ and $\delta^{15}N$), **d** illite crystallinity (standard deviations were calculated using three independent sets) and calibrated dates, **e** holocene temperature history of Antarctic Peninsula modified from Mulvaney et al. [7], and **f** X-ray diffractograms of clay minerals (S: smectite; I: illite; C: chlorite; K: kaolinite).

proximal to the grounding zone of ice shelf during the LGM[30], whereas the overlying characteristic laminations of U2–3, for example, yellowish brown sandy mud (21–40 cm) and rhythmic couplets (~6 couplets/cm) of laminated silt and clay (120–180 cm) are typical of sedimentary facies when an ice shelf retreats and undergoes ice-thinning process during the Holocene climate optimum[31]. Finally, the presence of IRD and foraminifera in the bioturbated sediment of U1 support oxic and seasonally opened marine conditions[10]. X-ray diffraction (XRD) profiles show that the major mineral compositions for the clay size sediments throughout the core are smectite, chlorite, kaolinite, and illite and lepidocrocite (Supplementary Figs. 1 and 2). Depth profiles of clay minerals throughout the core show that illite is dominant (50–60%) compared with smectite (~10 %), chlorite (~20%), and kaolinite (~15%). There is a clear separation of chlorite (14 Å) and smectite (17.5 Å) for glycolated samples (Supplementary Figs. 1 and 2).

Negative values of $\delta^{15}N$ within U4 (Fig. 2c) suggest that no organic matter was supplied from the marine environment to these sediments, more typical of a glacial till. In contrast, abrupt increase of $\delta^{15}N$ within U3, coupled with consistently lighter values of $\delta^{13}C$ through U4 and U3 and a complete lack of diatom frustule or fragments[29], suggest a stable sub-ice shelf environment apparently controlled by proximity to the grounding line of LIS[32]. Similar values of $\delta^{13}C$ between U4 and U3 likely reflect the

contribution of the suspended particles transported from the grounding line. Higher in the core, the increase in both $\delta^{13}C$ and $\delta^{15}N$ up to the values observed in U2 and U1 (U1: −25.7‰ and 3.8‰; U2: −25.9‰ and 3.2 ‰; see Supplementary Table 1) is likely to be the result of gradually increasing supply of the marine organic matter from open water column as reflected in grain-size distribution of illite and the presence of diatom frustule fragments in U1[24].

**Evidence for alteration of illite**. Alteration of illite minerals can be expressed in terms of IC (°Δ2θ)[16]. The average values of IC in Core EAP13-GC15B show variation with depth in the core (Fig. 2d, f). Importantly, an abrupt decrease in IC values from ~1.0 (U3) to <0.8 (U2–1) was observed during U2 (at ~4000 BP) (Fig. 2e), suggesting less-altered illite close to the surface. Overall, alteration of the illite between U3 and U1 is also supported by the modification of the illite structure from randomly ordered/ diffused Bragg's reflections in U3 to a discrete ordered pattern in U1 (Fig. 3), and variation in the Fe-oxidation state of the illite particles, with more Fe(III) present at shallower depths (35% in U1 and 17% in U3; Fig. 4a).

Such variability in IC might be caused by changes in the composition or supply of illite to the core. Histograms of the grain-size distribution of illite within the core do flatten, broaden,

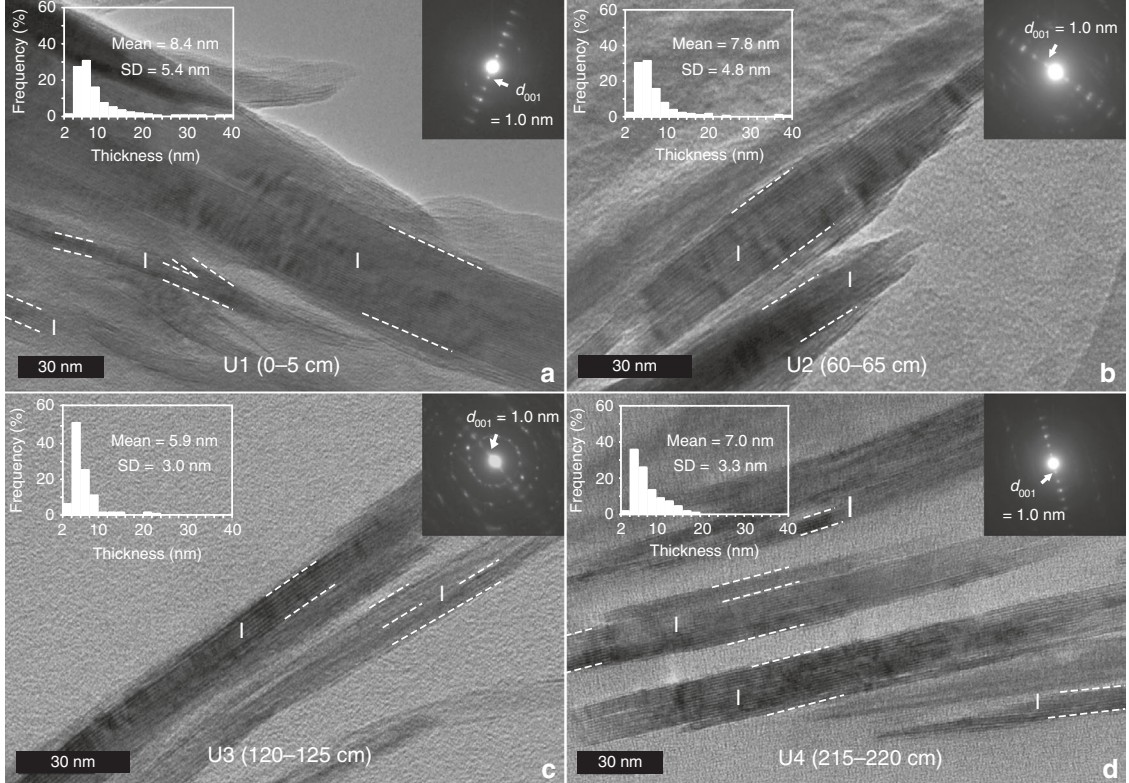

**Fig. 3 Transmission electron microscopy micrographs with selected area diffraction patterns and grain-size distribution.** Samples with depths of **a** 0–5 cm (U1), **b** 60–65 cm (U2), **c** 120–125 cm (U3), and **d** 215–220 cm (U4) of the core EAP13-GC16B (dashed lines show the boundary of illite (I) packets). The inset histograms of the grain-size distribution of illite (U4–1) flatten, broaden, and shift to larger size with a high standard deviation. The inset selected area electron diffractions of illite ($d_{001} = 1.0$ nm) shows modification of illite structure of randomly ordered/diffused Bragg's reflections (U3) to discrete ordered pattern (U1).

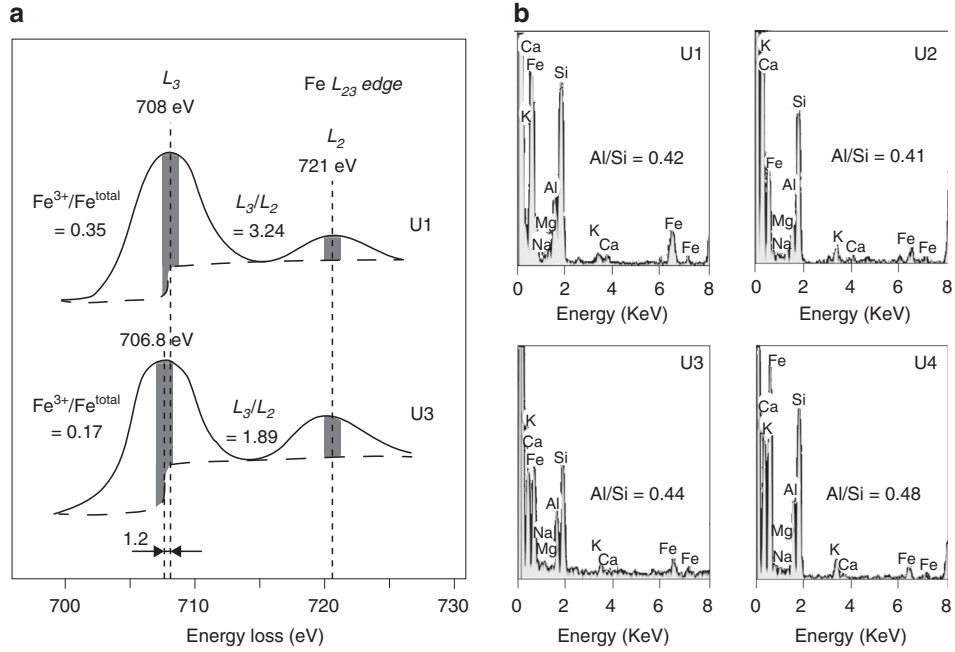

**Fig. 4 Fe-oxidation states and elemental composition of illite. a** Electron energy loss spectroscopy spectra of illite grains for U1 and U3 measuring a chemical shift of Fe $L_3$ edges (~1.2 eV). A calculated integral intensity ratio of Fe $L_3/L_2$ (3.24 and 1.89) for illite corresponds to $Fe^{3+}/Fe^{total}$ of 35 and 17% for U1 and U3, respectively. **b** Elemental composition typical of illite with 9–14 wt% of Fe contents and 0.41–0.44 of Al/Si for U1–3 compared to 6 wt% of Fe and 0.48 of Al/Si for U4.

and shift to a slightly larger grain size with a high standard deviation (SD = 5.4) within U1 compare to U2–4 (Fig. 3). However, a similar elemental composition of illite is observed throughout U1–3 with a higher content of Fe (9–14 wt%) compared to U4 (6 wt%) and an Al/Si ratio of illite of 0.41–0.44 during U1–3, compared with 0.48 for U4 (Fig. 4b). There is no discernible appearance of detrital muscovite (Al/Si ≈ 1), biotite (Al/Si ≈ 0.3), or paragonite (Al/Si ≈ 1)[33] that could affect the values of IC[34] (Fig. 4b). Moreover, measured values of $IC_{air\ (median)} - IC_{gly\ (median)}$ at each depth (0.14–0.52) correspond to the maximum 3% difference in smectite contents in illite/smectite mixed layers (I/S)[34,35] (Supplementary Table 2). The variation in IC for the air-dried and ethylene-glycolated samples showed a similar trend with increasing depth, suggesting that the effect of smectite contents in I/S on IC is minimal in this study (Supplementary Fig. 3). Also, rare-earth element composition indicates that Holocene sediments (U1–3) are of the same source, which is different from the source at the LGM (U4) (Supplementary Fig. 4). These observations suggest that variations in IC during the Holocene cannot be explained simply by different sources of illite minerals or mineralogical variation, but rather by in situ alteration.

Previous observations of such diagenetic mineral alteration generally are reported in high temperature and pressure environments[36], far from the conditions observed in sub-ice shelf sediments. Thus, we suggest here that biogeochemical redox-sensitive reactions, possibly by iron-reducing bacteria rather than high temperature and pressure, are the likely cause of the modification of the illite structure observed in U3. Microbial Fe reduction in 2:1 layered phyllosilicate structure results in the alteration of net negative charge, crystal lattice energy, cation exchange capacity, and distribution of Fe(II)–Fe(III) in the octahedral sheet[37–39] that could modify the crystal domain size and structure[39] responding in the IC[16]. Transmission electron microscopy (TEM) measurements on the bioreduced illite confirmed the alteration of illite structure through reductive dissolution and decrease in illite crystalline size[24]. Indeed, smaller illite domain packet size displayed where reducing condition is favored (Fig. 3), measuring a high value of half-height width of illite (high values in IC) (Fig. 2d). Other clay minerals may also undergo microbially induced changes that involve release of reduced iron; however, illite is the only clay mineral for which we can currently measure the crystallinity responding to alteration of crystal structure in various redox conditions[25,40]. Furthermore, the consequences of sediment oxygenation with the onset of full open marine conditions in U1 are clearly revealed by the decrease of IC and a change in selected area electron diffraction (SAED) patterns of diffused, randomly ordered (U3) to discrete reflections (U1) (Fig. 3a, c), as illite alteration reactions cease.

**Evidence for microbial alteration and implications.** Significant positive correlations between the distribution/abundance of putative Fe-reducing bacteria and IC were observed in the core ($r = 0.90$ and 0.69, $p < 0.05$, Fig. 5). The major groups of bacteria present were identified as members of *Comamonadaceae* sp. (class *Betaproteobacteria*) and *Desulfobacteraceaet* sp. (class *Deltaproteobacteria*)[22,23], comprising only minor components of the microbial community in the top of the core, but becoming abundant below 50 cm (mid-U2) (Fig. 5a). The genomic signatures (e.g., gene contents) supporting the presence of iron-reducing metabolisms in the family Comamonadaceae in our samples were not found from the microbial genome database in the GenBank (https://www.ncbi.nlm.nih.gov/genome), indicating the occurrence of novel member with not-yet-known functions in those families. Nonetheless, the previous study[22]

suggested that the family Comamonadaceae is capable of reducing Fe in sediments (Fig. 5b). Furthermore, a recent study[41] showed a possibility of anaerobic ammonium oxidation coupled to ferric Fe reduction by this family. The correlation between *Desulfobacteraceae* and illite alteration (Fig. 5c) may reflect either direct microbial Fe reduction[42] or microbially influenced reductive dissolution of Fe-bearing minerals by hydrogen sulfide formed by sulfate-reducing bacteria[43,44]. Moreover, it was reported that Fe can be an electron acceptor during the reoxidation of $H_2S$[43]. The distribution of *Dehalococcoidetes* sp. (phylum *Chloroflexi*) may reflect a tight coupling with Fe-reducing bacteria[45]; however, they are not considered to be a direct driver to change IC (Fig. 5d). Given the low temperatures at these sediment depths compared to where IC is normally observed, and the observed relationship between the presence of putative Fe-reducing bacteria and IC, we hypothesize that microbial activity is responsible for the alteration of illite observed in these sediments, although the exact mechanism of electron transfer by which this occurs is not yet constrained[46,47]. Nevertheless, the biogeochemical data described here indicate a clear link between microbial Fe respiration, alteration of illite crystallinity, and an increase in the Fe(II) content of illite under anoxic conditions. The relative amount of putative Fe-reducing bacteria diminishes abruptly in the sediments under occasionally open marine conditions (U1), where laminated sediment structures have been destroyed by oxygenation and resultant epibenthic community development. Although such psychrophilic Fe reduction of illite results in similar structural and chemical modification (Figs. 3 and 4) to what has been presented in previous studies of mesophilic[13] and thermophilic reaction[48], our data are the first observations in natural sediments to suggest a possible pathway for Fe reduction (and release to sediment porewaters as dissolved Fe) associated with microbial Fe respiration at low temperature[21] in Antarctic regions[19].

**Sediment-derived Fe in the antarctic.** Sediments as an important or even dominant Fe source for the surface oceans has gained traction in recent years, challenging the primacy of atmospheric supply[49]. This growing consensus of the importance of submarine sediment-derived Fe sources in fueling surface productivity has been driven both by models[50] and by observations by the International GEOTRACES program, which have indicated transport of sediment-derived iron over $10^6$ km in the open ocean[51], including the Southern Ocean[52]. In the Antarctic, a number of recent studies have focused on the local importance of Fe from shelf sediments around Antarctica, suggesting that Fe sourced from sediments on the West Antarctic Peninsula is an important Fe source to coastal waters[53–55]. Sediment-derived Fe is also thought to be the dominant Fe source fueling blooms in the Ross and Amundsen Seas[56,57], and near Antarctic island chains[58].

The isotope signature of Fe is generally lighter at the depths of the Antarctic Intermediate Water in the Atlantic Sector of the Southern Ocean and S. Atlantic[59,60], consistent with the possibility of long-distance transport of sediment-derived Fe of a reductive origin[61,62] on the Antarctic Peninsula. Such an observation adds weight to the idea that microbial Fe(III) reduction in Fe-rich sediments on the Antarctic margins could play an important role in driving primary productivity in the Southern Ocean, especially near Antarctica. However, dissimilatory reductive dissolution of sediment release isotopically light $Fe^{2+}$ is typically thought to involve Fe(III) minerals such as goethite, hematite, and ferrihydrite[63,64]. Here, we propose that illite may provide an additional substrate for microbial reduction that was previously thought to be largely inaccessible. This study opens the door for iron isotope studies in the future aiming to

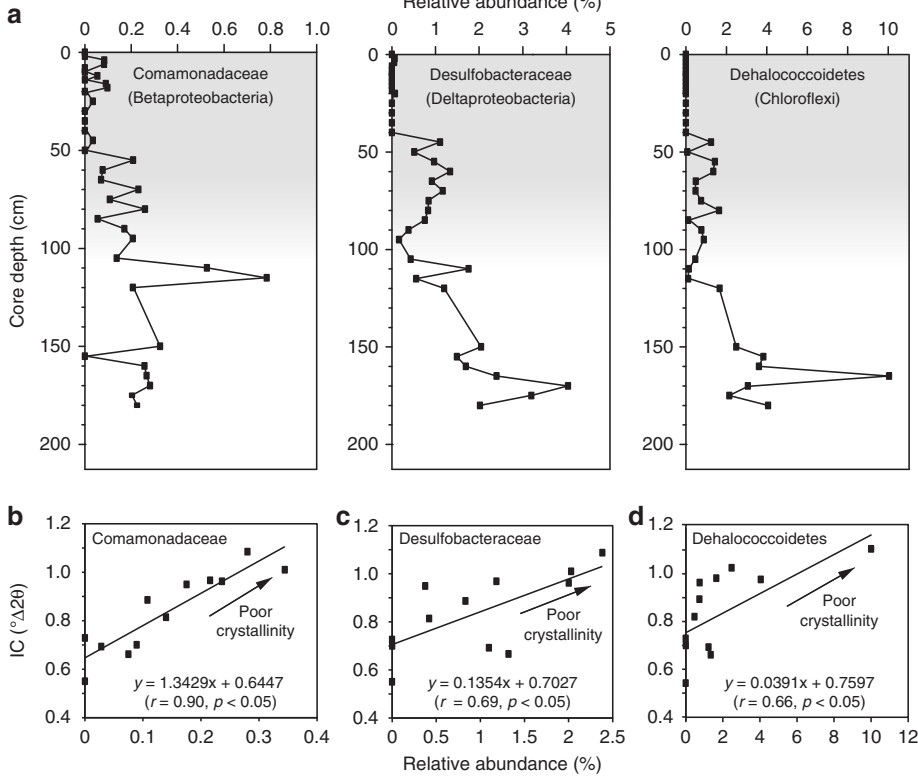

**Fig. 5 Distribution and abundance of putative Fe-reducing bacteria with illite crystallinity. a** Distribution of major groups in core EAP13-GC16B with depths (shaded areas suggest that oxic condition with occasionally open sea). Correlations of bacterial groups **b** *Comamonadaceae*, **c** *Desulfobacteraceae*, and **d** *Dehalococcoidetes* with illite crystallinity (IC).

constrain the proportion of iron release from Fe(III) oxides as well as illite by this process if there is a signature isotope composition for this process. This may be especially important in the Antarctic, because illite (and clay minerals more generally) appears so prominently on the continental shelf around Antarctica[14], and because redox dynamics are likely to be so intrinsically linked to ice shelf dynamics. Clay minerals comprise 10–30% of bulk Antarctic sediment[14], and our experimental data suggest that ~4% of Fe may be released by psychrophilic microbial reduction of the clay mineral smectite (Supplementary Fig. 5). Together, these observations suggest that microbial alteration of Fe-bearing clay minerals corresponding to redox conditions in deglacial sediments may also play a role in generating Fe(II) in sediment porewaters, which could potentially be released to the water column. The strength of this flux is likely to change with ice shelf conditions.

**Conceptual model for microbially enhanced Fe release from illite.** Given the potential importance of the amount of Fe(II) that can be sourced from the microbe–mineral interactions, we have observed that it is important to consider this as a changing source of Fe(II) to the Southern Ocean as ice shelves retreat[4]. Our opportunistic core indicates that changes in ice shelf position and oxygenation of the core top can create a boundary to diffusive/ advective export of sedimentary Fe(II) that forms from microbially driven IC downcore (Fig. 6). This process can also switch the flux of porewaters from the sediment from mainly diffusive to advective as the sediment is disturbed. Thus, the observed bioturbation at the surface of the core (Fig. 6a), although independent from continued microbially driven IC changes (Fig. 6b, c), will likely decrease the amount of Fe(II) sourced from the sediment. However, oxygenation of the surface sediments by bioturbation and bioirrigation as the ice shelf recedes or collapses

might increase the advective or resuspended flux of iron from the sediments[65,66]. Thus, any feedback between ice shelf retreat and $CO_2$ in the atmosphere would have to consider changes in both the form and quantity of iron coming from sub-ice shelf sediments. Between 2010 and 2016, the continent of Antarctica lost between 672 and 2254 $km^2$ of grounded ice area[9]. Given the prominence of illite in Antarctic sediments, coupled with the likely decrease in Fe(II) diffusion to the water column, as these sediments are oxygenated and exposed to more vigorous open-ocean exchange (Fig. 6a), such accelerated loss of ice shelf-covered seafloor is likely to dramatically alter both the flux and speciation of dissolved Fe reaching the ocean, providing a new possible feedback between ice-shelf dynamics (Fig. 6b–e) and primary productivity in the Southern Ocean.

## Methods
**Geological location and sediment sampling**. Sediment cores from the LIS-C embayment are exceedingly rare. The region is often inaccessible to ocean-going and ice-breaking research vessels, and as a result, has not been explored geologically to the extent of the lower latitude embayments of LIS-A and LIS-B. A marine geological expedition (ANA03C Cruise Expedition by the Korea Polar Research Institute (KOPRI)) was conducted in the LIS-A, LIS-B, and LIS-C embayments of northwestern Weddell Sea in 2013 (Fig. 1). LIS-C, the largest ice shelf in the Antarctic Peninsula (AP), has persisted since the LGM, but it has been thinning since the recent regional warming of AP[8]. A whole round core sample (238 cm below the seafloor) of marine sediment (EAP13-GC16B) was collected on the northwestern part of LIS-C embayment (66° 3.898′S, 60° 27.692′ W, Fig. 1). The sediments underneath the sea ice in front of the LIS-C had not been exposed until a recent retreat of the calving front[67]. This core is unique because it contains a record of the complete depositional features during the Holocene without modifications. For these reasons, our core represents an excellent test for the capability of biogeochemical reactions to alter mineralogical characteristics through the Holocene. The Larsen Ice Shelf system is a climate-sensitive glacial system; the loss of other Larsen ice shelves (LIS-A, 1995; LIS-B, 2002) along the AP is one of the most dramatic environmental changes directly observed anywhere on Earth[68].

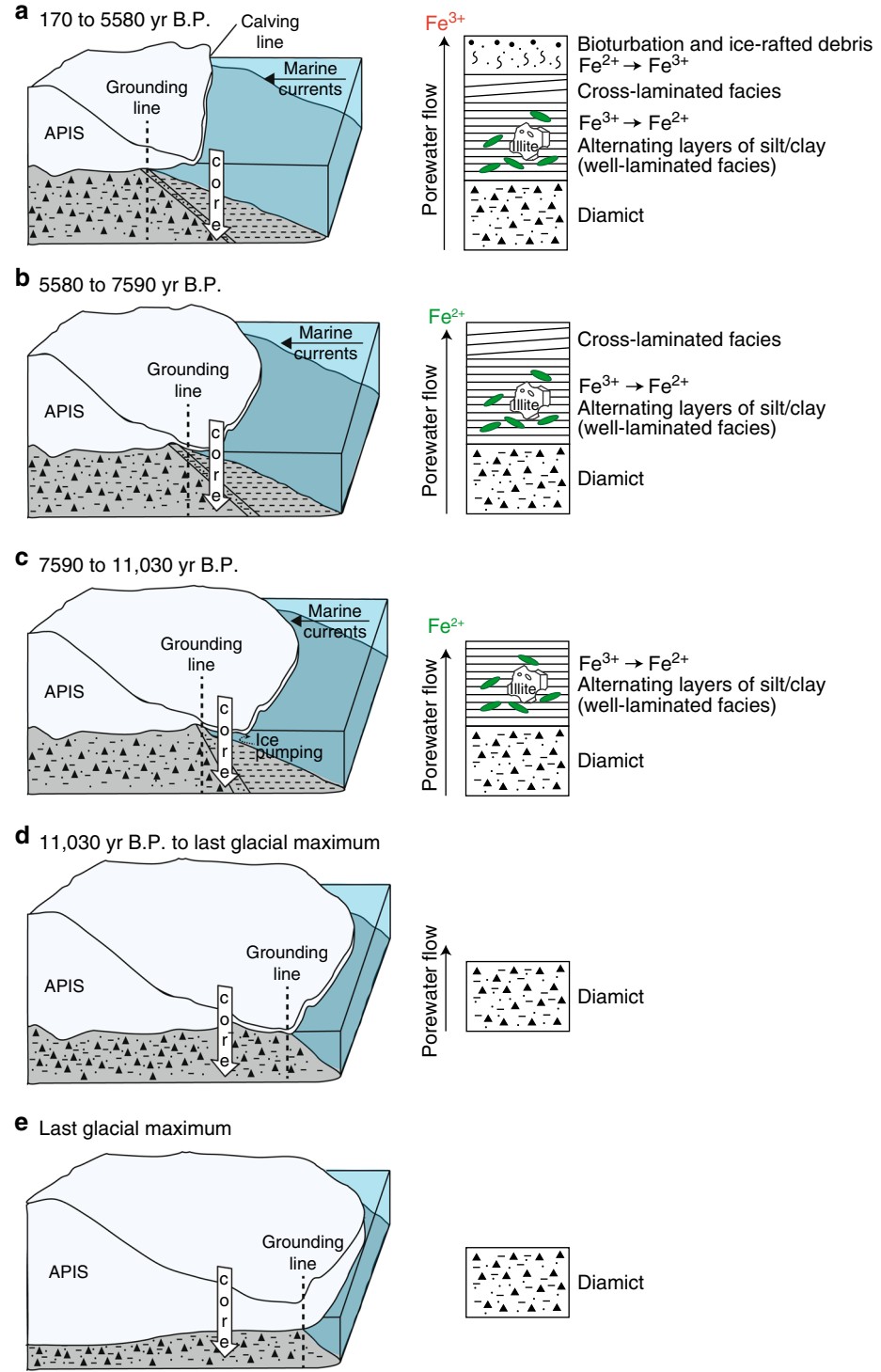

**Fig. 6 Conceptual model of Larsen Ice Shelf C retreat and microbe–mineral interaction in sediments.** The core site experienced five different depositional settings. **a** Oxygenated and bioturbated sediment in open marine (170 to 5580 yr BP), **b** cCross-laminated layers over laminated sediment under sub-ice shelf (5580 to 7590 yr BP), **c** laminated sediment under sub-ice shelf (7590 to 11,030 yr BP), **d** diamict under subglacial (11,030 yr BP to Last Glacial Maximum), and (**e**) diamict in Last Glacial Maximum (APIS: Antarctic Peninsula Ice Sheet).

**Petrophysical properties**. A half slab of 238 cm whole round core was examined by X-ray radiograph to investigate the sedimentary structures. The working half of the core was sampled every 4-cm depth interval for measurement of size fractions of clay (<2 µm), silt (>2 to <62.5 µm), and sand (>62.5 µm) following a conventional particle separation procedure[69]. Briefly, particles larger than 62.5 µm (gravel and sand) were separated from bulk sediment by wet sieving. For the fine fractions smaller than 62.5 µm (silt and clay), particles were settled down in a 1 L graduated cylinder for 3 days, and then the grain size was determined using Micrometrics

Sedigraph 5100. Qualitative ordinal color scale (Munsell color chart) was used to define bedding distinct color variations in the core.

**Isotopic composition**. Forty-four subsamples (~50 mg of each sample) were collected into tin capsules to analyze isotopic composition of $\delta^{13}C$ and $\delta^{15}N$ in sediment with depths. Each sample was pretreated with 1 M of hydrochloric acid three times in order to completely remove carbonates, and then isotopic

compositions with a precision of 0.2‰ were measured at the Stable Isotope Laboratory, GNS Science, Lower Hutt, New Zealand, using an Isoprime isotope ratio mass spectrometer, interfaced to an EuroEA elemental analyzer in continuous-flow mode (EA-IRMS).

**Electron microscopy**. TEM micrographs for the sediment with depths were recorded using JEOL JEM-2100F at Ewha Women's University, Seoul. The TEM samples were prepared following the impregnation procedure for LR White resin[70] to minimize the confusion of selecting 10 Å illite packets from the collapsed hydrous clay minerals, such as smectite, displaying the same spacings under the high-energy TEM beam. Illite layers were confirmed by SAED patterns with the strongest Bragg reflections of 1.0, 0.5, and 0.33 nm. The TEM specimens were then sectioned with 700 Å in thickness using a diamond knife microtome (ULTRACUT TCT; Leica, installed at the Korea Basic Science Institute, KBSI). Transmission electron microscopy-electron energy loss spectroscopy (TEM-EELS) was also applied to quantify the oxidation states of Fe in illite structure as a function of the integral ratio of Fe $L_3/L_2$[71] using a TECHNAI F30 ST TEM at the KBSI, Seoul. The operational conditions for EELS acquisition were an energy dispersion of 0.1 eV/channel, entrance aperture of 2.0 mm, and the full-width at half-maximum to 1.0 eV for the zero-loss peak calibration. The statistical optimum signal-window parameters for the integral ratio of Fe-$L_{2,3}$ edges were calculated using the Gatan Inc.'s Digital Micrograph™ software. The backgrounds were removed from EELS spectra by Double arctan functions and Standard power law[71].

**X-ray diffractometer**. XRD analyses were performed on air-dried clay sample (<2 μm) with each depth at a scan speed of 1°/min with a Rigaku Miniflex II automated diffractometer utilizing Cu-Kα radiation at Yonsei University, Seoul. Clay size fraction samples were dispersed in distilled water (0.7 mg/mL) and put in an ultrasonic water bath for 30 s to prevent flocculation of particles. Then, air-dried samples were made by pipetting sediment dispersions onto slide glasses for XRD analysis[72]. IC, also known as Kübler index, comprises the half-height width of illite 10-Å peak from XRD profiles[73] utilizing the Search-Match and OriginPro8 software after the background was removed by Chebyshev polynomial with ≤20 coefficients, and the pseudo-Voigt profile function proposed by Thompson et al.[74]. IC was originally referred to as the Weaver index[75] that reflects the X-ray scattering domain size and structural distortions[16], measuring the crystal alterations[76]. Three independent sets of XRD profiles were collected in order to reduce the errors in the measurements. Semiquantitative evaluations of clay minerals were measured after the glycolation treatment. The relative percentage of each clay mineral was calculated using weighting factors[77].

**DNA extraction, microbial community composition**. Genomic DNA was extracted from sediment samples every 2 cm from top to 20 cm and every 5 cm below 20 cm (~0.5 g of each sample) using the FastDNA Spin Kit for soil (MP Biomedicals). The quantity of genomic DNAs was measured by Picogreen fluorometry. Polymerase chain reactions (PCRs) for the 16S ribosomal RNA (rRNA) genes of prokaryote (Bacteria and Archaea) were performed using primers Uni787F and Uni1391R[78]. We adopted the PCR cycling conditions according to Jorgensen et al. [78], except for applying the nested PCR, in order to minimize PCR bias. PCR amplicon pyrosequencing sequences were processed using the QIIME software package, ver. 1.8[79]. First, raw flowgram data were filtered and denoized by the AmpliconNoise software, version 1.29[80], using the platform option for FLX titanium sequence data implemented in QIIME. Sequences were clustered based on operational taxonomic units at 97% similarity using UCLUST[81] and classified using the 16S rRNA gene sequence reference database GreenGenes 13_8[82]. To avoid effects of different sample sizes for estimating diversity comparisons among prokaryotic communities, sequences were resampled to the smallest library[83].

**Bacterial culture and experimental procedure**. Psychrophilic bacteria (*Shewanella vesiculosa* 21939, 21996 and *Shewanella frigidimarina* 21881) known as cold-tolerant and cold-adapted facultative FeRB[46,84] were isolated from King George Island in western Antarctica by KOPRI. Psychrophilic bacteria were cultured in Luria–Bertani broth (LB) liquid medium aerobically for 7 days at 15 °C to increase cell concentration and activity. The culture was then washed three times with 0.1 mM NaCl solution to remove the residual LB medium, and the washed cells were re-incubated anaerobically for 7 days in liquid M1 medium[85] with Fe(III)-citrate (34.5 mM) as an electron acceptor and Na-lactate (20 mM) as an electron donor in order to adapt Fe respiration. The solution was buffered by MOPS (3-(N-morpholino)propanesulfonic acid) to maintain a pH of 7. The incubated cells were finally washed three times with 0.1 mM NaCl solution to remove residual Fe(III)-citrate and M1 medium. Cells were then inoculated in the liquid M1 medium with nontronite (5 g/L) in each set as an electron acceptor and Na-lactate (20 mM) as an electron donor. Control samples were treated identically to experimental ones, except no bacteria to compare the microbial effect on Fe reduction. A sample of 100 mL of this suspension in each serum bottle was placed in the incubator at 4 °C for up to 4 months.

**Reporting summary**. Further information on research design is available in the Nature Research Reporting Summary linked to this article.

## Data availability

The source data underlying figures and Supplementary Material are provided as a Supplementary Data file. Any other information that supports data interpretation is available from the corresponding authors upon reasonable request.

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

## Acknowledgements

The present research was supported by the National Research Foundation of Korea (NRF) grant funded by the Korea government (MSIP) (NRF-2018R1A2B6002036) and Antarctic Project of KOPRI (PE19030) to J.W.K. We thank the science parties and crews of KOPRI icebreaker R/V *Araon* for their extraordinary efforts to collect samples and data processing (KOPRI ANA03C). Especially, we express our deep appreciation to the late Dr. E. Domack and Dr. Yoon as co-chief scientists of Weddell Sea expedition for their tremendous contributions to this project.

## Author contributions

All authors contributed to the writing of the manuscript. J.K. designed the overall research concept, data interpretation, and drafted the manuscript. J.J. produced data, including TEM data process and microbial diversity information with the help of K.Y. and C.Y.H. K.-C.Y., H.I.Y., and J.I.L. interpreted the sedimentary facies and depositional environments. B.E.R. and C.S. interpreted the chronology of the core, and B.E.R. and T.M.C. developed the concept of a possible Fe source. B.E.R. and J.J. conceived and designed the conceptual model of porewater iron transitions through time.

## Competing interests

The authors declare no competing interests.
