## [Peer Review File · Nature Communications]

Reviewers' comments:

Reviewer #1 (Remarks to the Author):

This is a very nice study of the crystallinity of illite from cores collected in the Larsen Shelf C area. There appears to be a link between the crystallinity of the illite, the Fe content, the Eh of the sediment and the quantity and types of microbes found within the sediment. The hypothesis is that microbes are driving Fe loss from the illite, even though the temperatures are low. Folk think that this is an abiotic process elsewhere.

I very much like the data, the observations and I am sure that the authors are right that microbial processes result in the effects they observe.

My slight reservation is the process or mechanism by which the Fe is lost. How is it envisioned that microbes extract the Fe from the illite. My hunch is that the microbes are promoting the conditions that can result in Fe release from illites, but the precise mechanism by which this happens is something else. Microbes must be promoting similar conditions in the warmer sediments elsewhere that the authors highlight, but why then are these warmer processes seen as abiotic rather than microbially driven, and what is it about sub-ice shelf sediments or the biogeochemistry of the sediment environment that leads to microbial processes dominating?

I think the hook that microbes solve the problem in the cold - abiotic processes might be/would be too slow in these environments - is clever, but my gut feeling is that there is more to this story than is told at the moment.

Am I going to recommend publication? Of course. Great data set, provocative idea, calls for clarification in how Fe loss from illites in the spectrum of marine environments really occurs. Go for it.

Reviewer #2 (Remarks to the Author):

This paper deals with the potential for mobilization and delivery of dissolved (aqueous) ferrous iron (Fe(II)aq) derived from clay (illite) in the Larsen Ice Shelf to the Southern Ocean during the Holocene, and the implications of glacial retreat on this process. Input of Fe from ice shelves to what are generally thought to be Fe-limited Southern Ocean waters could have an important influence primary productivity, and possibly modulate the climate response to glaciation as ice shelves advance or retreat. To address this question, the authors obtained and characterized a marine sediment core from the northwestern part of the Larsen Ice Shelf C (LIS-C). The core encompassed four major lithological units, where the upper unit (U1) and the upper portion of the second unit (U2) are bioturbated, reflecting the presence of relatively oxidized bottom water conditions that arose as a result of ice shelf retreat something like 5000 years ago (5 ka). In contrast, the lower portion of U2 and the third unit (U3) are laminated, reflecting reducing bottom water conditions that were present as a result of relatively stagnant bottom water during the waning phase of the last glacial maximum ca. 5-12 ka. Among the various (solid-phase only; see below) measurements performed on the core, key for this paper was the analysis of down core changes in the crystallinity of the Fe-bearing clay mineral illite, which apparently comprises approximately 50% of the clay mineral content of Antarctic sub-ice shelf sediments. The authors put forward the hypothesis that alteration of illite coupled to the activity of dissimilatory iron-reducing bacteria (DIRB) could be responsible for release of Fe(II)aq from sub-ice shelf sediments during periods of prominent glaciation.

While this basic hypothesis generally makes sense, I am perplexed by the singular focus on illite in this paper. Although I do not doubt the validity of the downcore illite crystallinity analyses, in and of themselves these data do not prove that microbial illite transformation would have been the main source of Fe(II)aq released to Southern Ocean waters. It is well-known that Fe(III)-bearing clays such as smectite and illite, though subject to microbial reduction, generally do not undergo anything like quantitative dissolution during this process. In contrast, Fe(III) oxides generally

dissolve during microbial reduction (not completely, but much more so than clays), and it thus seems entirely possible that Fe(III) oxide reduction – rather than clay reduction – would be the more likely contributor to Fe(II)aq export from subglacial shelf sediments. Along these lines, I was surprised that the authors did not conduct standard wet-chemical extraction procedures to estimate Fe(III) oxide content through the downcore redox transition. Another set of analytical data that is missing is pore water Fe(II)aq concentrations; it seems to me that in order to argue convincingly for the past release of Fe(II)aq from unit U3 and the lower portion of U2, the authors should demonstrate that porewater in these units contain substantial amounts of Fe(II)aq. By analogy, porewater should be very low in Fe(II)aq in unit U1 and the upper portion of unit U2. To my opinion, without such data, together with information on other possible Fe(III)-bearing mineral phases that may contribute to Fe(II)aq generation, it is not valid to infer that microbial illite reduction would be the main source of Fe(II)aq release to the Southern Ocean.

I realize that the authors might rebut this conclusion by pointing out the correlation between putative (see final comment below) DIRB abundance and illite crystallinity shown in Fig. 4. Here again, however, while I don't doubt that the correlation is real, this does not prove that illite was the source of Fe(III) for sustenance of DIRB populations. It seems entirely possible that Fe(III) oxides (as well as perhaps smectite, which is generally more readily reduced by bacteria than illite) also served as electron acceptors for DIRB. In other words, even if illite was undergoing reductive transformation by DIRB, this doesn't prove that illite would be the dominant Fe(III) source for Fe(II)aq generation. Along analogous lines, without further information on the Fe mineralogy, the incubation experiment with psychrophilic DIRB (two *Shewanella* species isolated from the Antarctic Peninsula) shown in Fig. S2 cannot be used to definitively infer that illite reduction was the main source of Fe(II)aq release, as some or all of the Fe(II)aq could have come from Fe(III) oxide reduction. In summary, the authors need to provide substantial additional analysis and interpretation to properly address their hypothesis that illite reduction was a major source of Fe(II)aq release from Larsen Ice Shelf sediments during the Holocene.

Regarding the putative DIRB populations inferred from 16S rRNA gene amplicon sequencing (Fig. 4), none of the three families (Comamonadaceae, Dehalococcoidetes, and Desulfobacteraceae) identified are canonical DIRB taxa. Although some organisms within the Desulfobacteraceae have been shown to be capable of Fe(III) reduction (Lovley et al., *Mar. Geol.*, 113:41-53, 1994), this is by no means an established idea. And to my knowledge, neither the Dehalococcoidetes or Desulfobacteraceae families have been shown to contain organisms capable of Fe(III) reduction. So unless I'm missing something, the correlation between the abundance of these taxa and illite crystallinity could be completely spurious, having nothing to do with the role of these organisms as DIRB. In fact, given that the Desulfobacteraceae are well-known as sulfate-reducing bacteria, it seems possible that the correlation reflects the impact of hydrogen sulfide attack on illite as opposed to direct microbial reduction.

Reviewer #3 (Remarks to the Author):

In this manuscript, Jung and colleagues examine a marine sediment core on the northwestern part of the Larsen Ice Shelf C (LIS-C) embayment. The authors argued that the microbe-mineral interactions may release ferrous iron (Fe²⁺) to porewaters from illite. Although this paper can be of potential interest for readers of *Nature communication*, I cannot recommend the manuscript for publication in its present form.

This is an important investigation of the possible sources of Fe²⁺ in the Southern Ocean. However, the conclusion is not fully supported by the results presented in this paper. The key statement that "The biogeochemical data described here indicates a clear link between microbial Fe respiration, alteration of illite crystallinity and an increase in the Fe(II) content of illite under anoxic conditions." (Line 156-158) is vague. The biogeochemical data presented include X-ray diffraction of clay fraction, TEM characterization of illite particles and abundance of Fe reducing bacteria. In my opinion, the link extracted from the correlations from the depth profiles is weak due to

following reasons:

- 1) The depth profiles of illite crystallinity might reflect the variation of smectite contents in illite/smectite mixed-layer components. The authors mentioned the presence of smectite in the core materials (Line 104), however, was not able to discuss the nature of the smectite and how it may impact the measurement of illite crystallinity. The existence of illite/smectite mixed-layer minerals in surface sediments in the Antarctic Ocean has been extensively documented in literature. The amount of smectite is likely to have a greater impact on the illite crystallinity rather than possible microbial modification. As shown in Fig. 1 in the manuscript, the smectite peaks at 14 \AA are higher at depth where the illite crystallinity is higher (e.g. 136-190 cm).
- 2) The depth profiles of abundance of Fe reducing bacteria might reflect the variation of environmental factors such as decreasing concentrations of dissolved oxygen (DO) with depth.
- 3) The link between abundance of Fe reducing bacteria and illite crystallinity. Illite is not the only Fe-bearing minerals in the core. Even we are convinced that the microbial Fe reduction occurs, how do we know the bacteria utilize illite but not chlorite or Fe oxyhydroxide? Are there any data showing the depth profiles of chlorite and Fe oxyhydroxide?

More specific comments:

Line 57-59: Be more specific about the alteration. I think in metamorphic the alteration usually means smectite-illite transition. Do you mean the same?

Line 72: there are several papers about illite reduction by microorganisms (Dong et al., EST, 2003; Zhang et al., Chemical Geology, 2012).

Line 103-105: It is surprising that no quantitative XRD data (on clay fraction) were presented here. Are XRD data on ethylene-glycolated samples available?

Line 193: There is little information in Supplementary. I didn't follow how the experiment was conducted and why smectite was mentioned here? Very unclear.

Fig. 3: not sure how the diffraction pattern was collected? Please explain the dashed, white line and short white bar.

Reviewers' comments:

Reviewer #1 (Remarks to the Author):

This is a very nice study of the crystallinity of illite from cores collected in the Larsen Shelf C area. There appears to be a link between the crystallinity of the illite, the Fe content, the Eh of the sediment and the quantity and types of microbes found within the sediment. The hypothesis is that microbes are driving Fe loss from the illite, even though the temperatures are low. Folk think that this is an abiotic process elsewhere. I very much like the data, the observations and I am sure that the authors are right that microbial processes result in the effects they observe.

→ We thank the reviewer for noting the value of our observations and ideas.

My slight reservation is the process or mechanism by which the Fe is lost. How is it envisioned that microbes extract the Fe from the illite. My hunch is that the microbes are promoting the conditions that can result in Fe release from illites, but the precise mechanism by which this happens is something else. Microbes must be promoting similar conditions in the warmer sediments elsewhere that the authors highlight, but why then are these warmer processes seen as abiotic rather than microbially driven, and what is it about sub-ice shelf sediments or the biogeochemistry of the sediment environment that leads to microbial processes dominating?

- We agree that the precise mechanism is not well constrained, although there is evidence from culture work. In general, the precise mechanisms by which iron-reducing microbes transfer electrons to insoluble minerals are unknown in nature, and require further study beyond the scope of this manuscript. Based on results from laboratory studies with pure cultures, *Geobacter* produces nanowires (i.e. pili) containing cytochromes which facilitate electron transfer to the surface of iron oxide minerals. *Shewanella* either produces similar electrically conductive appendages or uses riboflavins as electron shuttles, depending on culture conditions. Depending on the optimum temperature, microbes such as psychrophiles (<15 °C), mesophiles (20-45 °C), and thermophiles (>60 °C) showed different activities (Bozal et al., 2009; Jaisi et al., 2011; Lovley and Phillips, 1988; Zhang et al., 2007). To address this in the manuscript, we have added a clause to the sentence to make this clearer: **(Line163)** "...although the exact mechanism of electron transfer by which this occurs is not yet constrained (Bozal et al., 2009; Jaisi et al., 2011; Lovley and Phillips, 1988)." However, despite the exact mechanism being unconstrained, we would argue that the data presented here provide compelling evidence for the influence of microbial activity as reviewer commented.

I think the hook that microbes solve the problem in the cold - abiotic processes might be/would be too slow in these environments - is clever, but my gut feeling is that there is more to this story than is told at the moment. Am I going to recommend publication? Of course. Great data set, provocative idea, calls for clarification in how Fe loss from illite in the spectrum of marine environments really occurs. Go for it.

- We agree with the reviewer, and anticipate that our provocative study will spur on more work to identify the exact mechanism and systematics for the illite breakdown process, now that it has been identified in Antarctic sediments at low temperatures and pressures. This modification is conventionally thought to be affected by temperature and pressure in diagenetic environment (Eberl and Hower, 1976; Freed and Peacor, 1992).

Reviewer #2 (Remarks to the Author):

This paper deals with the potential for mobilization and delivery of dissolved (aqueous) ferrous iron (Fe(II)aq) derived from clay (illite) in the Larsen Ice Shelf to the Southern Ocean during the Holocene, and the implications of glacial retreat on this process. Input of Fe from ice shelves to what are generally thought to be Fe-limited Southern Ocean waters could have an important influence primary productivity, and possibly modulate the climate response to glaciation as ice shelves advance or retreat. To address this question, the authors obtained and characterized a marine sediment core from the northwestern part of the Larsen Ice Shelf C (LIS-C). The core encompassed four major lithological units, where the upper unit (U1) and the upper portion of the second unit (U2) are bioturbated, reflecting the presence of relatively oxidized bottom water conditions that arose as a result of ice shelf retreat something like 5000 years ago (5 ka). In contrast, the lower portion of U2 and the third unit (U3) are laminated, reflecting reducing bottom water conditions that were present as a result of relatively stagnant bottom water during the waning phase of the last glacial maximum ca. 5-12 ka. Among the various (solid-phase only; see below) measurements performed on the core, key for this paper was the analysis of down core changes in the crystallinity of the Fe-bearing clay mineral illite, which apparently comprises approximately 50% of the clay mineral content of Antarctic sub-ice shelf sediments. The authors put forward the hypothesis that alteration of illite coupled to the activity of dissimilatory iron-reducing bacteria (DIRB) could be responsible for release of Fe(II)aq from sub-ice shelf sediments during periods of prominent glaciation.

While this basic hypothesis generally makes sense, I am perplexed by the singular focus on illite in this paper.

Although I do not doubt the validity of the downcore illite crystallinity analyses, in and of themselves these data do not prove that microbial illite transformation would have been the main source of Fe(II)aq released to Southern Ocean waters. It is well-known that Fe(III)-bearing clays such as smectite and illite, though subject to microbial reduction, generally do not undergo anything like quantitative dissolution during this process. In contrast, Fe(III) oxides generally dissolve during microbial reduction (not completely, but much more so than clays), and it thus seems entirely possible that Fe(III) oxide reduction – rather than clay reduction – would be the more likely contributor to

Fe(II)aq export from subglacial shelf sediments. Along these lines, I was surprised that the authors did not conduct standard wet-chemical extraction procedures to estimate Fe(III) oxide content through the downcore redox transition. Another set of analytical data that is missing is pore water Fe(II)aq concentrations; it seems to me that in order to argue convincingly for the past release of Fe(II)aq from unit U3 and the lower portion of U2, the authors should demonstrate that porewater in these units contain substantial amounts of Fe(II)aq. By analogy, porewater should be very low in Fe(II)aq in unit U1 and the upper portion of unit U2. To my opinion, without such data, together with information on other possible Fe(III)-bearing mineral phases that may contribute to Fe(II)aq generation, it is not valid to infer that microbial illite reduction would be the main source of Fe(II)aq release to the Southern Ocean.

- We thank the reviewer for noting the validity of the illite changes and the logic of our hypothesis and for giving us the opportunity to clarify our thinking and focus on illite, compared to other Fe-bearing minerals. As the reviewer points out, the possible source of Fe induced by microbial Fe(III) reduction would be a range of Fe-oxides, and Fe-bearing clay minerals including smectite, illite, and chlorite (Dong et al., 2003b; Jaisi et al., 2007; Kostka et al., 1999; Zhang et al., 2012). Indeed, several papers have shown that Fe-oxides are the major source of Fe-liberation associated with Fe-respiration (Bhatia et al., 2013; Canfield et al., 1993; Liu et al., 2007; Monien et al., 2014; Raiswell et al., 2008). This is well known. However, the novel aspect of our paper is instead the identification of illite breakdown as a new potential and important Fe source, which all of the reviewers recognized as novel and worthy of publication. We did not intend to say that it was the only or perhaps even the major Fe-source mineral in sedimentary environments, and we apologize if this was unclear. We would also argue that inclusion of other minerals would only enhance the iron source, and would likely be influenced by ice sheet changes in similar ways to illite. We do comment on other iron minerals in the paper. We have also added more interpretation of XRD data showing 4 possible Fe sources such as lepidocrocite, smectite, chlorite, and illite. We modified in the abstract **(Line 18)**: “...illite and other Fe-bearing minerals.....”. We added new data in the text **(Line102-109)**: “X-ray diffraction profiles show that the major mineral composition for the clay size sediments throughout the core is smectite (S), chlorite (C), kaolinite (K), and illite (I) and lepidocrocite (L) (Fig. 2 and Supplementary Fig. S3). Depth profiles of clay minerals (Supplementary Fig. S4) throughout the core shows that illite is dominant (50-60 %) compared with smectite (~10 %), chlorite (~20 %), and kaolinite (~15 %). There is a clear separation of chlorite (14 Å) and smectite (17.5 Å) for the glycolated samples and no XRD peak for illite/smectite interstratified layer (9.84 degree 2-theta) was observed. (Supplementary Fig. S3)”. We also added in the text **(Line 200)**: “.....clay structures.....” Previous work (Jaisi et al., 2005; Kim et al., 2010; Kostka et al., 1999; Urrutia et al., 1998), has shown that the amount of Fe-release depends on mineralogy, surface charge, particle size, and crystal chemistry. It is well known that Fe-oxides are the major source of Fe (Hawkings et al., 2014; Turner and Hunter, 2001), while clay minerals also showed Fe-release associated with microbial Fe-reduction. In the text we also added other Fe sources from Fe-oxides **(Line 66-68)**: “Whereas a range of iron minerals are known to be sources of dissolved Fe upon breakdown by iron-reducing or -oxidizing bacteria (Emerson et al., 2015; Reyes et al., 2016), adding illite to this group, and at low temperatures,” We also added in the text **(Line 144-147)**: “Other clay minerals may also undergo microbial-induced changes that involve release of reduced iron, however illite is the only clay mineral for which we can currently measure the crystallinity responding to alteration of crystal structure in various redox conditions (Dong et al., 2003a; Jaisi et al., 2007; Kostka et al., 1999; Zhang et al., 2012).” It is practically impossible to quantify the amount of Fe-release from the mixture of Fe-bearing minerals (Fe-oxides and clay minerals), particularly for the natural sediments, and we do not attempt to do that here. In the paper by Kostka et al. (1999), crystalline magnetite showed less microbial Fe-reduction compared to the smectite. Nonetheless, goethite and amorphous Fe showed a large extent of Fe(III) reduction (Kostka et al., 1999). Again, we did not show that illite is the major source of Fe(II) release to Southern Ocean water. We suggest that IC could be an indicator of depositional conditions under retreat and advance of Ice Shelf. Based on the IC, we can infer that Fe-bearing minerals including illite should undergo the same redox-reaction with microbes, releasing Fe(II), that responds to the movement of ice-shelf. Please note that we addressed the objective of our paper in the text: “To address the possibility of illite crystallinity changes sourcing Fe to the water column beneath an ice

shelf,...(Line 71)" We also addressed the importance of Fe-oxides minerals as a Fe-source in the text: "However, dissimilatory reductive dissolution of sediment releases isotopically light Fe²⁺ is typically thought to involve Fe(III) minerals such as goethite, hematite and ferrihydrite⁵⁸⁻⁶⁰. Here, we propose that illite may also provide a substrate for microbial reduction that was previously thought to be largely inaccessible. This may be especially important in the Antarctic, since illite, and clays more generally, appear so prominently on the continental shelf around Antarctica¹⁴...(Line 191-196)".

Regarding the reviewer's surprise that we did not analyze porewaters, we agree with the reviewer's comments on the strength that porewater analysis would have added to this study. Unfortunately, the cores were not originally collected for clay mineralogy analysis, and, as Reviewer 1 pointed out, the results are so novel as to be unanticipated by the authors at the onset of this work. Thus, porewater samples were not taken for iron analysis. Now that the cores have been in the repository for several years, we do not feel like this type of analysis would reveal anything meaningful – it is simply too late to resample and obtain results which are representative. Our results, however, will ensure that future cores taken from beneath Antarctic ice shelves and elsewhere on the continental margin will likely be sampled in this way – by our group and others working on similar problems. This will be the impact of this paper once published.

I realize that the authors might rebut this conclusion by pointing out the correlation between putative (see final comment below) DIRB abundance and illite crystallinity shown in Fig. 4. Here again, however, while I don't doubt that the correlation is real, this does not prove that illite was the source of Fe(III) for sustenance of DIRB populations.

- ➔ Again, we did not suggest that illite is the only source of Fe-release, and we apologize if this was not made clear enough in our original submission. By quantifying oxidation states of Fe in illite by EELS acquisitions, IC corresponds to the redox conditions and microbial activity (Of course, other Fe-bearing minerals such as lepidocrocite, smectite, and chlorite undergo the same conditions as illite). However, illite is the only mineral that we can measure the crystallinity responding to alteration of crystal structure in various redox conditions. In the present study area, neither pressure nor heat can be the factor that modify the illite structure. Furthermore, the population of FeRB is inversely related with the value of IC. We addressed in the text the possible factors to control IC in the conventional diagenetic settings : "Previous observations of such diagenetic mineral alteration generally are reported in high temperature and pressure environments³⁴, far from the conditions observed in sub-ice shelf sediments. Thus, here, instead of high temperature and pressure, we suggest that biogeochemical redox-sensitive reactions,...(Line 140)"

It seems entirely possible that Fe(III) oxides (as well as perhaps smectite, which is generally more readily reduced by bacteria than illite) also served as electron acceptors for DIRB. In other words, even if illite was undergoing reductive transformation by DIRB, this doesn't prove that illite would be the dominant Fe(III) source for Fe(II)aq generation. Along analogous lines, without further information on the Fe mineralogy, the incubation experiment with psychrophilic DIRB (two *Shewanella* species isolated from the Antarctic Peninsula) shown in Fig. S2 cannot be used to definitively infer that illite reduction was the main source of Fe(II)aq release, as some or all of the Fe(II)aq could have come from Fe(III) oxide reduction. In summary, the authors need to provide substantial additional analysis and interpretation to properly address their hypothesis that illite reduction was a major source of Fe(II)aq release from Larsen Ice Shelf sediments during the Holocene.

- ➔ We did measure smectite, chlorite, illite and lepidocrocite for Fe-bearing minerals in our natural sample, and this information has been added to the paper. As we cited previous studies (Dong et al., 2003b; Jaisi et al., 2007), and remarked above, we did not state that illite is the major or only source of Fe-release in our study area. What is new about this study is that we show that illite could be another additional source, complementing other Fe sources. Furthermore, illite crystallinity can be related to depositional conditions and thus potentially act as proxy for other sedimentary sources of Fe.

Regarding the putative DIRB populations inferred from 16S rRNA gene amplicon sequencing (Fig. 4),

none of the three families Comamonadaceae, Dehalococcoidetes, and Desulfobacteraceae) identified are canonical DIRB taxa. Although some organisms within the Desulfobacteraceae have been shown to be capable of Fe(III) reduction (Lovley et al., Mar. Geol., 113:41-53, 1994), this is by no means an established idea. And to my knowledge, neither the Dehalococcoidetes or Desulfobacteraceae families have been shown to contain organisms capable of Fe(III) reduction. So unless I'm missing something, the correlation between the abundance of these taxa and illite crystallinity could be completely spurious, having nothing to do with the role of these organisms as DIRB. In fact, given that the Desulfobacteraceae are well-known as sulfate-reducing bacteria, it seems possible that the correlation reflects the impact of hydrogen sulfide attack on illite as opposed to direct microbial reduction.

- Fe-reducing bacteria are phylogenetically diverse across phyla. Conventionally the characteristic of Fe reduction has been confirmed for pure bacterial cultures like *Geobacter* and *Shewanella*. However, taking into account that cultivatable bacteria generally comprise less than 0.1-1% of natural bacterial community, previously known Fe-reducing bacteria found by cultivation-dependent approaches are underrepresented so far. Recently a combination of geochemical measurements and cultivation-independent molecular techniques (e.g. next-generation sequencing) revealed the presence of new potential Fe-reducing bacterial groups in sediment environments. The family Comamonadaceae (Betaproteobacteria) and *Desulfobacter* which is a type genus of the family Desulfobacteraceae (Deltaproteobacteria) were presumed to be putative Fe-reducing bacteria in recent studies (Emerson et al., 2015; Reyes et al., 2016). We have added the literature for referring to those groups as putative Fe-reducing bacteria in the revised manuscript. The distribution of Dehalococcoidetes reflects a tight-coupling with Fe-reducing bacteria, however they are not a direct driver to change IC (Wei and Finneran, 2011). We modified the caption to figure 4 and added text (Line 155-160): "The major groups of bacteria present were identified as *Comamonadaceae* sp. (class *Betaproteobacteria*) and *Desulfobacteraceae* sp. (class *Deltaproteobacteria*)^{33,34}, comprising only minor components of the microbial community in the top of the core, but becoming abundant below 50 cm (mid-U2). The distribution of *Dehalococcoidetes* sp. (phylum *Chloroflexi*) also reflected a tight-coupling with Fe-reducing bacteria³⁵, however they are not considered to be a direct driver to change IC (Fig. 4d)." Figure 4 caption (Line 559): "*Comamonadaceae* (b), and *Desulfobacteraceae* (c) with IC. A positive correlation of the abundance of Fe-reducing bacteria to the increase in IC is measured with the correlation coefficient values ($r= 0.90$, and 0.69). The distribution of *Dehalococcoidetes* (d) reflected a tight-coupling with Fe-reducing bacteria, however they are not a direct driver to change IC."

Reviewer #3 (Remarks to the Author):

In this manuscript, Jung and colleagues examine a marine sediment core on the northwestern part of the Larsen Ice Shelf C (LIS-C) embayment. The authors argued that the microbe-mineral interactions may release ferrous iron (Fe^{2+}) to porewaters from illite. Although this paper can be of potential interest for readers of Nature communication, I cannot recommend the manuscript for publication in its present form.

This is an important investigation of the possible sources of Fe^{2+} in the Southern Ocean. However, the conclusion is not fully supported by the results presented in this paper. The key statement that “The biogeochemical data described here indicates a clear link between microbial Fe respiration, alteration of illite crystallinity and an increase in the Fe(II) content of illite under anoxic conditions.” (Line 156-158) is vague. The biogeochemical data presented include X-ray diffraction of clay fraction, TEM characterization of illite particles and abundance of Fe reducing bacteria. In my opinion, the link extracted from the correlations from the depth profiles is weak due to following reasons:

- We appreciate the reviewer’s appraisal of the importance of our investigation and the constructive criticism that will make this manuscript a driver of iron biogeochemistry research in the Southern Ocean. We address the reviewer’s criticisms point-by-point below.

1) The depth profiles of illite crystallinity might reflect the variation of smectite contents in illite/smectite mixed-layer components. The authors mentioned the presence of smectite in the core materials (Line 104), however, was not able to discuss the nature of the smectite and how it may impact the measurement of illite crystallinity. The existence of illite/smectite mixed-layer minerals in surface sediments in the Antarctic Ocean has been extensively documented in literature. The amount of smectite is likely to have a greater impact on the illite crystallinity rather than possible microbial modification. As shown in Fig. 1 in the manuscript, the smectite peaks at 14 Å are higher at depth where the illite crystallinity is higher (e.g. 136-190 cm).

- We added XRD profiles for the glycolated samples with depths as the reviewer 2 requested (Supplementary Figure S3). Illite is dominant (50 – 60 %) compared with smectite contents (around 7-10%) through the depths. At depths between 136-190cm, XRD peaks with 14 Å d-spacing is not smectite as you can clearly separate chlorite (14 Å) and smectite (17.5 Å) for the glycolated samples. Thus, there would be no smectite effects on the IC in the present samples. Illite/smectite interstratified layer has not been observed in our samples. There are no peaks at 9.84 degree 2-theta throughout the depths. We added in the text (Line 102-109): “X-ray diffraction profiles show that the major mineral composition for the clay size sediments throughout the core is smectite (S), chlorite (C), kaolinite (K), and illite (I) and lepidocrocite (L) (Fig. 2 and Supplementary Fig. S3). Depth profiles of clay minerals (Supplementary Fig. S4) throughout the core shows that illite is dominant (50-60 %) compared with smectite (~10 %), chlorite (~20 %), and kaolinite (~15 %). There is a clear separation of chlorite (14 Å) and smectite (17.5 Å) for the glycolated samples and no XRD peak for Illite/smectite interstratified layer (9.84 degree 2-theta) was observed. (Supplementary Fig. S3)”.

Figure captions S3 and S4 were added: Supplementary Figure S3. X-ray diffraction (XRD) patterns of air-dried and glycolated clay (<2 µm) in sediment core from site EAP13-GC17 at various depths (S: smectite, C: chlorite, K: kaolinite, I: illite, Pl: plagioclase, Q: quartz, L: lepidocrocite). There is a clear separation of chlorite (14 Å) and smectite (17.5 Å) for the glycolated samples. Supplementary Figure S4. Depths profiles of clay minerals in the EAP13-GC16B core. The depths profiles of clay minerals shows that illite is dominant (50-60%) compared with smectite (~10%), chlorite (~20%), and kaolinite (~15%).

2) The depth profiles of abundance of Fe reducing bacteria might reflect the variation of environmental factors such as decreasing concentrations of dissolved oxygen (DO) with depth.

- Although a depth profile of DO was not determined, oxygen penetration depth (OPD) likely would not exceed 50-60 cm in our sediment core (Sachs et al., 2009). In our results, a variation of Fe-reducing bacteria was relatively small in the upper oxic layer (indicated sedimentologically as bioturbated sediment), while the variation was large in the lower anoxic layer (Fig. 4, indicated sedimentologically as laminated, undisturbed sediment). Thus, DO is

not likely to be an environmental factor for explaining the abundance of Fe-reducing bacteria in our study.

3) The link between abundance of Fe reducing bacteria and illite crystallinity. Illite is not the only Fe-bearing minerals in the core. Even we are convinced that the microbial Fe reduction occurs, how do we now the bacteria utilize illite but not chlorite or Fe oxyhydroxide? Are there any data showing the depth profiles of chlorite and Fe oxyhydroxide?

- As we discussed for the previous reviewer's comments about the possible source of Fe, in the present sediment we measured Fe-bearing minerals including smectite, illite, chlorite, and lepidocrosite by XRD. Of course, we did not intend to say illite is the major source of Fe, associated with microbial activity. However, we would say variations in IC can be recorded by microbial Fe-reduction in illite structure implying the sediment depositional conditions associated with the retreat and advance of ice shelf. Our EELS data acquisition indicated that variation in Fe-oxidation states in illite is closely linked to the microbial activity under retreat and advance of Ice Shelf. Again, please note our responses to other possible Fe source above (See our responses to Reviewer 2).

More specific comments:

Line 57-59: Be more specific about the alteration. I think in metamorphic the alteration usually means smectite-illite transition. Do you mean the same?

- As we described (Line 140), IC can determine the degree of diagenetic states of pelitic rock for the low-grade metamorphism, by measuring half height width of illite XRD peak that corresponds to the illite alteration with high temperature and pressure. It is surprising to detect the illite alteration given the variation in the redox conditions and microbial activity in the glacial-interglacial periods. REE data and crystal size distribution indicate that illite clay minerals characterized with IC in the core are from the same source. Therefore, alteration in our study must be associated with factors other than conventional ones (temperature and pressure). Microbial activity showed linear correlation with IC with depth, indicating that Fe-reduction in illite structure is related to the modification of illite structure that responds in IC.

Line 72: there are several papers about illite reduction by microorganisms (Dong et al., EST, 2003; Zhang et al., Chemical Geology, 2012).

- Yes, we included those published papers (Line 69) that discussed the microbial Fe-reduction in various clay minerals. Because we have illite, smectite, chlorite and lepidocrosite we discussed the possible source of Fe from clay minerals as well as Fe-oxides. Nonetheless, crystalline phase of magnetite showed less Fe-reduction rate than clay minerals, suggesting that there are several factors such as particle size, crystal chemistry, structure, and concentration of clays or Fe-oxides that control the Fe-release from the minerals. It is not easy to say which mineral can contribute more Fe-release as pointed out by Dong et al (2003). Again, we did not intend to say illite is the only source of Fe-release associated with microbial activity in the present sediment. Previous papers have shown a strong relationship between microbial Fe-reduction of clay minerals and Fe-release (Zhang et al., 2012). Furthermore, we showed microbially induced Fe-release from biogenic smectite at low temperature, suggesting the feasibility of alteration of clay minerals (Supplementary Fig.S2) along with Fe-oxides in Antarctic area.

Line 103-105: It is surprising that no quantitative XRD data (on clay fraction) were presented here. Are XRD data on ethylene-glycolated samples available?

- We added XRD profiles showing relative contents of clay minerals with increasing depth (Supplementary Figure S3 and S4). As we discussed above, illite is the dominant phase while smectite is minor content in the sediments. We added in the text **(Line 102-109): "X-ray diffraction profiles show that the major mineral composition for the clay size sediments throughout the core is smectite (S), chlorite (C), kaolinite (K), and illite (I) and lepidocrocite (L)**

(Fig. 2 and Supplementary Fig. S3). Depth profiles of clay minerals (Supplementary Fig. S4) throughout the core shows that illite is dominant (50-60 %) compared with smectite (~10 %), chlorite (~20 %), and kaolinite (~15 %). There is a clear separation of chlorite (14 Å) and smectite (17.5 Å) for the glycolated samples and no XRD peak for Illite/smectite interstratified layer (9.84 degree 2-theta) was observed. (Supplementary Fig. S3)".

Line 193: There is little information in Supplementary. I didn't follow how the experiment was conducted and why smectite was mentioned here? Very unclear.

- In response to the reviews of this manuscript, we performed more batch experimentation with clays and microbes at low temperature. As we discussed above, previous papers showed the Fe-reduction in various clay minerals such as illite, chlorite and smectite indicating that Fe-release can be from various origins. In this manuscript, we showed one example of Fe-release from clay minerals associated with microbial Fe-reduction at low temperature for the first time, suggesting the feasibility of microbial role in Fe-release from clay minerals at low temperature. We provided the experimental setting in the method (Bacterial culture and experimental procedure) of the supplementary information file. Our experimental data (Supplementary Figure S2) suggests ~4 % of Fe may be released by psychrophilic microbial reduction of the clay mineral smectite (nontronite). Of course, this is single mineral experiment to understand the microbial Fe-reduction in clay minerals at low temperature. Therefore, we used nontronite which measured a large amount of structural Fe (23.4 % total Fe content by weight, where 99.4 % of the total Fe is Fe(III) (Jaisi et al., 2005) for the most optimum conditions of the microbial Fe-reduction at low temperature.

Fig. 3: not sure how the diffraction pattern was collected? Please explain the dashed, white line and short white bar.

- First, we verified the illite packet by measuring EDS and layer spacings with 10 Å. Then, illite layers were confirmed by Selected Area Electron Diffraction (SAED) patterns with the strongest Bragg reflections of 1.0, 0.5, and 0.33 nm (Fig. 3 inset). Dashed white line separated each illite packet. "A short white bar" is referred to the illite (I). We added in the text (Line 261-264): "... 10 Å illite packets from the collapsed hydrous clay minerals, such as smectite, displaying the same spacings under the high-energy TEM beam. Illite layers were confirmed by Selected Area Electron Diffraction (SAED) patterns with the strongest Bragg reflections of 1.0, 0.5, and 0.33 nm."
- In addition to the major changes, we changed or modified the words in the text to make it clear: Line111 added "more"; Line 132 corrected "for" to "within"; Line 187-199 deleted "Furthermore", added the "possibility"; Line 241 added "directly"

- Bhatia, M. P., Kujawinski, E. B., Das, S. B., Breier, C. F., Henderson, P. B., and Charette, M. A., 2013, Greenland meltwater as a significant and potentially bioavailable source of iron to the ocean: *Nature Geoscience*, v. 6, no. 4, p. 274.
- Bozal, N., Montes, M. J., Miñana-Galbis, D., Manresa, A., and Mercadé, E., 2009, *Shewanella vesiculosa* sp. nov., a psychrotolerant bacterium isolated from an Antarctic coastal area: *International journal of systematic and evolutionary microbiology*, v. 59, no. 2, p. 336-340.
- Canfield, D. E., Thamdrup, B., and Hansen, J. W., 1993, The anaerobic degradation of organic matter in Danish coastal sediments: iron reduction, manganese reduction, and sulfate reduction: *Geochimica et Cosmochimica Acta*, v. 57, no. 16, p. 3867-3883.
- Dong, H., Kostka, J. E., and Kim, J., 2003a, Microscopic evidence for microbial dissolution of smectite: *Clays and Clay Minerals*, v. 51, no. 5, p. 502-512.
- Dong, H., Kukkadapu, R. K., Fredrickson, J. K., Zachara, J. M., Kennedy, D. W., and Kostandarithes, H. M., 2003b, Microbial reduction of structural Fe (III) in illite and goethite: *Environmental Science & Technology*, v. 37, no. 7, p. 1268-1276.
- Eberl, D., and Hower, J., 1976, Kinetics of illite formation: *Geological Society of America Bulletin*, v. 87, no. 9, p. 1326-1330.
- Emerson, D., Scott, J. J., Benes, J., and Bowden, W. B., 2015, Microbial iron oxidation in the arctic tundra and its implications for biogeochemical cycling: *Appl. Environ. Microbiol.*, v. 81, no. 23, p. 8066-8075.
- Freed, R. L., and Peacor, D. R., 1992, Diagenesis and the formation of authigenic illite-rich I/S crystals in Gulf Coast shales: TEM study of clay separates: *Journal of Sedimentary Research*, v. 62, no. 2.
- Froelich, P. N., Klinkhammer, G., Bender, M. L., Luedtke, N., Heath, G. R., Cullen, D., Dauphin, P., Hammond, D., Hartman, B., and Maynard, V., 1979, Early oxidation of organic matter in pelagic sediments of the eastern equatorial Atlantic: suboxic diagenesis: *Geochimica et cosmochimica acta*, v. 43, no. 7, p. 1075-1090.
- Hawkings, J. R., Wadham, J. L., Tranter, M., Raiswell, R., Benning, L. G., Statham, P. J., Tedstone, A., Nienow, P., Lee, K., and Telling, J., 2014, Ice sheets as a significant source of highly reactive nanoparticulate iron to the oceans: *Nature communications*, v. 5.
- Jaisi, D. P., Dong, H., and Liu, C., 2007, Influence of biogenic Fe (II) on the extent of microbial reduction of Fe (III) in clay minerals nontronite, illite, and chlorite: *Geochimica et Cosmochimica Acta*, v. 71, no. 5, p. 1145-1158.
- Jaisi, D. P., Eberl, D. D., Dong, H., and Kim, J., 2011, The formation of illite from nontronite by mesophilic and thermophilic bacterial reaction: *Clays and Clay Minerals*, v. 59, no. 1, p. 21-33.
- Jaisi, D. P., Kukkadapu, R. K., Eberl, D. D., and Dong, H., 2005, Control of Fe (III) site occupancy on the rate and extent of microbial reduction of Fe (III) in nontronite: *Geochimica et Cosmochimica Acta*, v. 69, no. 23, p. 5429-5440.
- Kim, K., Choi, W., Hoffmann, M. R., Yoon, H.-I., and Park, B.-K., 2010, Photoreductive dissolution of iron oxides trapped in ice and its environmental implications: *Environmental science & technology*, v. 44, no. 11, p. 4142-4148.
- Kostka, J. E., Wu, J., Nealson, K. H., and Stucki, J. W., 1999, The impact of structural Fe (III) reduction by bacteria on the surface chemistry of smectite clay minerals: *Geochimica et Cosmochimica Acta*, v. 63, no. 22, p. 3705-3713.
- Liu, C., Zachara, J. M., Foster, N. S., and Strickland, J., 2007, Kinetics of reductive dissolution of hematite by bioreduced anthraquinone-2, 6-disulfonate: *Environmental science & technology*, v. 41, no. 22, p. 7730-7735.
- Lovley, D. R., and Phillips, E. J., 1988, Novel mode of microbial energy metabolism: organic carbon oxidation coupled to dissimilatory reduction of iron or manganese: *Applied and environmental microbiology*, v. 54, no. 6, p. 1472-1480.
- Monien, P., Lettmann, K. A., Monien, D., Asendorf, S., Wöflf, A.-C., Lim, C. H., Thal, J., Schnetger, B., and Brumsack, H.-J., 2014, Redox conditions and trace metal cycling in coastal sediments from the maritime Antarctic: *Geochimica et Cosmochimica Acta*, v. 141, p. 26-44.
- Raiswell, R., Benning, L. G., Tranter, M., and Tulaczyk, S., 2008, Bioavailable iron in the Southern Ocean: the significance of the iceberg conveyor belt: *Geochemical transactions*, v. 9, no. 1, p. 7.
- Reyes, C., Dellwig, O., Dähnke, K., Gehre, M., Noriega-Ortega, B. E., Böttcher, M. E., Meister, P., and Friedrich, M. W., 2016, Bacterial communities potentially involved in iron-cycling in Baltic Sea and North Sea sediments revealed by pyrosequencing: *FEMS microbiology ecology*, v. 92, no. 4, p. fiw054.

- Sachs, O., Sauter, E. J., Schlüter, M., van der Loeff, M. M. R., Jerosch, K., and Holby, O., 2009, Benthic organic carbon flux and oxygen penetration reflect different plankton provinces in the Southern Ocean: *Deep Sea Research Part I: Oceanographic Research Papers*, v. 56, no. 8, p. 1319-1335.
- Turner, D. R., and Hunter, K. A., 2001, *The biogeochemistry of iron in seawater*, Wiley Chichester, UK.
- Urrutia, M., Roden, E., Fredrickson, J., and Zachara, J., 1998, Microbial and surface chemistry controls on reduction of synthetic Fe (III) oxide minerals by the dissimilatory iron-reducing bacterium *Shewanella alga*: *Geomicrobiology Journal*, v. 15, no. 4, p. 269-291.
- Wei, N., and Finneran, K. T., 2011, Influence of ferric iron on complete dechlorination of trichloroethylene (TCE) to ethene: Fe (III) reduction does not always inhibit complete dechlorination: *Environmental science & technology*, v. 45, no. 17, p. 7422-7430.
- Zhang, G., Dong, H., Kim, J., and Eberl, D., 2007, Microbial reduction of structural Fe³⁺ in nontronite by a thermophilic bacterium and its role in promoting the smectite to illite reaction: *American Mineralogist*, v. 92, no. 8-9, p. 1411-1419.
- Zhang, J., Dong, H., Liu, D., Fischer, T. B., Wang, S., and Huang, L., 2012, Microbial reduction of Fe (III) in illite–smectite minerals by methanogen *Methanosarcina mazei*: *Chemical Geology*, v. 292, p. 35-44.

Reviewers' comments:

Reviewer #2 (Remarks to the Author):

I think that this is good, provocative science that will lead to further high profile work. I remain very positive in my support.

Reviewer #3 (Remarks to the Author):

I must acknowledge that the authors did their best to respond to my comments on the original manuscript. However, on balance the changes/additions did very little to alter my take on the unwarranted focus on microbial illite transformation on Fe(II)aq release to Southern Ocean waters during Holocene glaciation, as well as other issues related to the microbial community analyses. As the authors readily admit, Fe(III) oxides could have been much more important sources of Fe(II) mobilization in these sediments. Unfortunately they chose **not** to conduct standard wet-chemical extraction procedures on the core materials to estimate Fe(III) oxide abundance so as to put some constraints their potential importance (or non-importance) as substrates for dissimilatory microbial reduction, e.g. in relation to the estimated abundance of illite and other clay minerals. I understand that the authors (as well as some of the reviewers) are excited about the idea of illite as a substrate for microbial reduction and Fe(II) mobilization, but this idea alone does not seem sufficient to justify publication of the paper in Nature Communications. In another words, I am not comfortable with the take home message being "microbial illite reduction **might** have been a source of Fe(II)aq mobilization in Holocene Southern Ocean sediments". Without further direct proof or constraints on this idea, the paper simply does not live up to the message conveyed by its title. In light of these weaknesses, in my opinion the only option to make the paper more realistic and honest would be to change the title to something like "Microbial Fe(III) reduction as a potential Fe source responding to depositional environments under the Larsen Ice Shelf C during the Holocene".

Regarding the 16S rRNA gene amplicon-based inferences of downcore dissimilatory Fe(III)-reducing bacterial abundances, my initial concerns still stand and the speculation that organisms from the *Comamonadaceae* and *Dehalococcoidetes* families are in fact Fe(III)-reducers in these sediments remains highly speculative. Also, I was surprised and disappointed that the authors did not bring forward the possibility that the correlation between *Desulfobacteraceae*-related 16S rRNA gene abundance and illite alteration reflects the impact of hydrogen sulfide attack on illite as opposed to direct microbial reduction.

Reviewer #4 (Remarks to the Author):

This is the manuscript I reviewed before. In my opinion, the manuscript has been improved. However, I think the authors need to discuss more about illite crystallinity (IC) and how the microbial reduction of illite may change IC.

As review by Abad (2007), there are several possible factors may affect IC, including domain size (crystallite size), lattice strain and the presence of other micaceous mineral phases with XRD peaks coincident with or adjacent to the illite reflection (such as detrital mica, paragonite, and illite/paragonite or illite/smectite mixed-layers). The domain size interpretation is consistent with the TEM images in Fig. 3. But I also notice that the values of IC changed significantly after ethylene glycol treatment. Such change is used by Jaboyedoff et al. (2001) to quantify the percentage of smectite layers in illite/smectite mixed-layers. I think the authors need to add more analysis and discussion on this. In my previous review, I already raised the concern on the illite/smectite mixed-layers. The authors replied as "no XRD peak for Illite/smectite interstratified layer (9.84 degree 2-theta) was observed." I don't really understand this reply.

The authors claimed that they “did not state that illite is the major or only source of Fe-release in our study area”. I am surprised about this statement because if illite reduction is not the major source of Fe, then what drives the change of the abundance of Fe reducing bacteria with depth. Do the authors infer that other Fe-bearing minerals are also reduced at depth? This need to be clarified.

References

Abad, I., 2007. Physical meaning and applications of the illite Kubler index: measuring reaction progress in low-grade metamorphism. In: Nieto, F., and Jiménez Millan, J.J. (Ed.), *Diagenesis and Low-Temperature Metamorphism: Theory, Methods and Regional Aspects*. Jaén, Union Grafica, Volume 3, p. 53-64.

M. Jaboyedoff, F. Bussy, B. Kübler, Ph. Thelin, 2001. ILLITE “CRYSTALLINITY” REVISITED. *Clays and Clay Minerals*; 49 (2): 156–167.

Reviewers' comments:

Reviewer #2 (Remarks to the Author):

I think that this is good, provocative science that will lead to further high profile work. I remain very positive in my support.

- Thanks again for the positive comments. We strongly agree with R2, because that is the main purpose of publication in the high impact journal like Nature Communications.

Reviewer #3 (Remarks to the Author):

I must acknowledge that the authors did their best to respond to my comments on the original manuscript. However, on balance the changes/additions did very little to alter my take on the unwarranted focus on microbial illite transformation on Fe(II)aq release to Southern Ocean waters during Holocene glaciation, as well as other issues related to the microbial community analyses. As the authors readily admit, Fe(III) oxides could have been much more important sources of Fe(II) mobilization in these sediments. Unfortunately they chose not to conduct standard wet-chemical extraction procedures on the core materials to estimate Fe(III) oxide abundance so as to put some constraints their potential importance (or non-importance) as substrates for dissimilatory microbial reduction, e.g. in relation to the estimated abundance of illite and other clay minerals. I understand that the authors (as well as some of the reviewers) are excited about the idea of illite as a substrate for microbial reduction and Fe(II) mobilization, but this idea alone does not seem sufficient to justify publication of the paper in Nature Communications. In another words, I am not comfortable with the take home message being “microbial illite reduction might have been a source of Fe(II)aq mobilization in Holocene Southern Ocean sediments”. Without further direct proof or constraints on this idea, the paper simply does not live up to the message conveyed by its title. In light of these weaknesses, in my opinion the only option to make the paper more realistic and honest would be to change the title to something like “Microbial Fe(III) reduction as a potential Fe source responding to depositional environments under the Larsen Ice Shelf C during the Holocene”.

- We understand the concern raised by R3, and we have modified the title as suggested. As R2 commented, we anticipate that this study will stimulate future high profile work, including the biotic mineral diagenesis under the ice shelves, Fe isotopic fractionation associated with biotic/abiotic reduction in various minerals including both clay minerals and Fe-oxides. We feel that the role of journals like Nature Communications are exactly why R2 is excited by the publication of these findings – it is to communicate potentially impactful multidisciplinary new ideas that are of significant interest to specialists in each field. Publication of this article will certainly lead to the type of wet-chemical techniques that R2 (and the coauthors) deem important, but that were not the original focus of research on the sediment cores collected.

Regarding the 16S rRNA gene amplicon-based inferences of downcore dissimilatory Fe(III)-reducing bacterial abundances, my initial concerns still stand and the speculation that organisms from the Comamonadaceae and Dehalococcoidetes families are in fact Fe(III)-

reducers in these sediments remains highly speculative. Also, I was surprised and disappointed that the authors did not bring forward the possibility that the correlation between Desulfobacteraceae-related 16S rRNA gene abundance and illite alteration reflects the impact of hydrogen sulfide attack on illite as opposed to direct microbial reduction.

- We thank the reviewer for bringing up this point again, and we agree that members of the family Dehalococcoidetes are not typically considered to be Fe(III)-reducers as suggested by Wei and Finneran et al. (2011). Nonetheless, the distribution of Dehalococcoidetes sp. (phylum Chloroflexi) vs. IC may reflect a tight-coupling with Fe-reducing bacteria (Wei and Finneran, 2011), although we agree that they are not considered to be a direct driver to change IC. To find genomic signatures (e.g. gene contents) supporting the presence of iron-reducing metabolisms in the family Comamonadaceae, we have searched Comamonadaceae bacteria closely-related to the sequences in our samples from the microbial genome database in the GenBank (<https://www.ncbi.nlm.nih.gov/genome>). There is no relevant data found in the database indicating the occurrence of novel member with not-yet-known functions in those families. Although we have no direct evidence in this study, the previous study (Emerson et al., 2015) suggested that the family Comamonadaceae is capable of reducing Fe in sediments. To address such concerns, we have toned down our statements by adding “putative” in the text. Also, we have added discussion of the possibility of Fe-reduction by hydrogen sulfide formed by sulfate-reducing bacteria (Desulfobacteraceae). We rewrote this part as below:

(Line 172-184) The genomic signatures (e.g. gene contents) supporting the presence of iron-reducing metabolisms in the family Comamonadaceae in our samples were not found from the microbial genome database in the GenBank (<https://www.ncbi.nlm.nih.gov/genome>) indicating the occurrence of novel member with not-yet-known functions in those families. Nonetheless, the previous study (Emerson et al., 2015) suggested that the family Comamonadaceae is capable of reducing Fe in sediments. Furthermore, recent study (Bao et al., 2017) showed a possibility of anaerobic ammonium oxidation coupled to ferric Fe reduction (AAOFe) by this family. The correlation between Desulfobacteraceae and illite alteration (Fig. 4c) may reflect either direct microbial Fe reduction (Enning and Garrelfs, 2014) or chemical microbially influenced reductive dissolution of Fe-bearing minerals by hydrogen sulfide formed by sulfate-reducing bacteria (Dos Santos Afonso and Stumm, 1992; Li et al., 2004). Moreover, it was reported that Fe can be an electron acceptor during the reoxidation of H₂S (Dos Santos Afonso and Stumm, 1992). The distribution of Dehalococcoidetes sp. (phylum Chloroflexi) may reflect a tight-coupling with Fe-reducing bacteria (Wei and Finneran, 2011), however they are not considered to be a direct driver to change IC (Fig. 4d).

Reviewer #4 (Remarks to the Author):

This is the manuscript I reviewed before. In my opinion, the manuscript has been improved.

- We thank the reviewer for noting the improvements in the manuscript in response to reviewers.

However, I think the authors need to discuss more about illite crystallinity (IC) and how the microbial reduction of illite may change IC. As review by Abad (2007), there are several possible factors may affect IC, including domain size (crystallite size), lattice strain and the presence of other micaceous mineral phases with XRD peaks coincident with or adjacent to the illite reflection (such as detrital mica, paragonite, and illite/paragonite or illite/smectite mixed-layers). The domain size interpretation is consistent with the TEM images in Fig. 3. But I also notice that the values of IC changed significantly after ethylene glycol treatment. Such change is used by Jaboyedoff et al. (2001) to quantify the percentage of smectite layers in illite/smectite mixed-layers. I think the authors need to add more analysis and discussion on this. In my previous review, I already raised the concern on the illite/smectite mixed-layers. The authors replied as “no XRD peak for Illite/smectite interstratified layer (9.84 degree 2-theta) was observed.” I don't really understand this reply.

→ We apologize for the confusion and not being clear enough. In response, we have thoroughly revised the ms with new supplementary data sets (Supplementary Figure S5 and Supplementary Table 2) that will address the factors that may control IC.

Supplementary Table 2. Illite crystallinity values from clay minerals in the <2 μm size fraction for the air-dried and glycolated specimens of the sediments from core EAP13-GC16B.

Supplementary Figure S5. Comparison of IC_{air} (air-dried samples) and IC_{gly} (ethylene glycolated samples) with increasing depth. Three independent measurements for each sample were made (Supplementary Table 2). Variation in the median values of IC with depths ($\text{IC}_{\text{air (median)}} - \text{IC}_{\text{gly (median)}}$) ranges 0.14-0.52 that corresponds to 5-8 % of smectite contents in illite/smectite mixed-layers (Abad et al., 2007; Jaboyedoff et al., 2001).

(Line 135-146) There is no discernible appearance of detrital muscovite ($\text{Al/Si} \approx 1$), biotite ($\text{Al/Si} \approx 0.3$) or paragonite ($\text{Al/Si} \approx 1$) (Dixon et al., 1990) that could affect the values of IC (Abad et al., 2007) (Fig. 3). Moreover, measured values of $\text{IC}_{\text{air (median)}} - \text{IC}_{\text{gly (median)}}$ at each depth (0.14 ~ 0.52) corresponds to the maximum 3 % difference in smectite contents in illite/smectite mixed-layers (I/S) (Abad et al., 2007; Jaboyedoff et al., 2001) (Supplementary Fig. S5). Indeed, the variation in IC for the air-dried and ethylene glycolated samples showed a very similar trend suggesting that the effect of smectite contents in I/S on IC is minimal (Supplementary Fig. S5). Also, rare earth element composition indicates that Holocene sediments (U1-3) are of the same source different from the LGM (U4) (Supplementary Fig. S1). These observations suggest that variations in IC during the Holocene cannot be explained simply by different sources of illite minerals or mineralogical variation, but rather by in situ alteration.

(Line 151-158) Microbial Fe-reduction in 2:1 layered phyllosilicate structure results in the alteration of net negative charge, crystal lattice energy, cation exchange capacity, and distribution of Fe(II)-Fe(III) in the octahedral sheet (Komadel et al., 2006; Sainz-Díaz et al., 2002; Stucki and Kostka, 2006) that could modify the crystal domain size, and crystal structure (Sainz-Díaz et al., 2002) responding in the IC (Eberl and Velde, 1989). TEM measurements on the bioreduced illite confirmed the alteration of illite structure through reductive dissolution and decrease in illite crystalline size (Dong et al., 2003b). Indeed, smaller illite domain packet size displayed where reducing condition is

avored (Fig. 3), measuring a high value of half-height width of illite (high values in IC) (Fig. 2).

The authors claimed that they “did not state that illite is the major or only source of Fe-release in our study area”. I am surprised about this statement because if illite reduction is not the major source of Fe, then what drives the change of the abundance of Fe reducing bacteria with depth. Do the authors infer that other Fe-bearing minerals are also reduced at depth? This need to be clarified.

→ This was the comments raised by Reviewer 2 in the first round of revision. We replied and modified the text as below in the round 1.

We thank the reviewer for noting the validity of the illite changes and the logic of our hypothesis and for giving us the opportunity to clarify our thinking and focus on illite, compared to other Fe-bearing minerals. As the reviewer points out, the possible source of Fe induced by microbial Fe(III) reduction would be a range of Fe-oxides, and Fe-bearing clay minerals including smectite, illite, and chlorite (Dong et al., 2003b; Jaisi et al., 2007; Kostka et al., 1999; Zhang et al., 2012). Indeed, several papers have shown that Fe-oxides are the major source of Fe-liberation associated with Fe-respiration (Bhatia et al., 2013; Canfield et al., 1993; Liu et al., 2007; Monien et al., 2014; Raiswell et al., 2008). This is well known. However, the novel aspect of our paper is instead the identification of illite breakdown as a new potential and important Fe source, which all of the reviewers recognized as novel and worthy of publication. We did not intend to say that it was the only or perhaps even the major Fe-source mineral in sedimentary environments, and we apologize if this was unclear. We would also argue that inclusion of other minerals would only enhance the iron source, and would likely be influenced by ice sheet changes in similar ways to illite. We do comment on other iron minerals in the paper. We have also added more interpretation of XRD data showing 4 possible Fe sources such as lepidocrocite, smectite, chlorite, and illite. We modified in the abstract **(Line 18): “...illite and other Fe-bearing minerals.....”**. We added new data in the text **(Line103-109): “X-ray diffraction profiles show that the major mineral composition for the clay size sediments throughout the core is smectite (S), chlorite (C), kaolinite (K), and illite (I) and lepidocrocite (L) (Fig. 2 and Supplementary Fig. S3). Depth profiles of clay minerals (Supplementary Fig. S4) throughout the core shows that illite is dominant (50-60 %) compared with smectite (~10 %), chlorite (~20 %), and kaolinite (~15 %). There is a clear separation of chlorite (14 Å) and smectite (17.5 Å) for the glycolated samples”**. Previous work (Jaisi et al., 2005; Kim et al., 2010; Kostka et al., 1999; Urrutia et al., 1998), has shown that the amount of Fe-release depends on mineralogy, surface charge, particle size, and crystal chemistry. It is well known that Fe-oxides are the major source of Fe (Hawkings et al., 2014; Turner and Hunter, 2001), while clay minerals also showed Fe-release associated with microbial Fe-reduction. In the text we also added other Fe sources from Fe-oxides **(Line 66-68): “Whereas a range of iron minerals are known to be sources of dissolved Fe upon breakdown by iron-reducing or -oxidizing bacteria (Emerson et al., 2015; Reyes et al., 2016), adding illite to this group, and at low temperatures,”** We also added in the text **(Line 158-161): “Other clay minerals may also undergo microbial-induced changes that involve release of reduced iron, however illite is the only clay mineral for which we can currently measure the crystallinity responding to alteration of crystal structure in various redox conditions (Dong et al., 2003a; Jaisi et al., 2007; Kostka et al., 1999; Zhang et al., 2012).”** It is practically impossible to quantify the amount of Fe-release from the mixture of Fe-bearing minerals (Fe-oxides and clay minerals), particularly for natural sediments, and we do not attempt

to do that here. In the paper by (Kostka et al., 1999), crystalline magnetite showed less microbial Fe-reduction compared to the smectite. Nonetheless, goethite and amorphous Fe showed a large extent of Fe(III) reduction (Kostka et al., 1999). Again, we did not show that illite is the major source of Fe(II) release to Southern Ocean water. We suggest that IC could be an indicator of depositional conditions under retreat and advance of Ice Shelf. Based on the IC, we can infer that Fe-bearing minerals including illite should undergo the same redox-reaction with microbes, releasing Fe(II), that responds to the movement of ice-shelf. Please note that we addressed the objective of our paper in the text: "To address the possibility of illite crystallinity changes sourcing Fe to the water column beneath an ice shelf,...(Line 72)" We also addressed the importance of Fe-oxides minerals as a Fe-source in the text: "However, dissimilatory reductive dissolution of sediment releases isotopically light Fe²⁺ is typically thought to involve Fe(III) minerals such as goethite, hematite and ferrihydrite (Beard et al., 1999; Crosby et al., 2007; Icopini et al., 2004). Here, we propose that illite may also provide a substrate for microbial reduction that was previously thought to be largely inaccessible. This may be especially important in the Antarctic, since illite, and clays more generally, appear so prominently on the continental shelf around Antarctica (Petschick et al., 1996)...(Line 203-209)". Furthermore, recognition of illite crystallinity changes that may be microbially-mediated opens the door for iron isotope studies in the future that may be able to constrain the proportion of iron released by this process if there is a signature isotope composition for this process.

Reference

- Abad, I., Nieto, F., and Millán, J., 2007, Physical meaning and applications of the illite Kübler index: measuring reaction progress in low-grade metamorphism: Diagenesis and Low-Temperature Metamorphism, Theory, Methods and Regional Aspects, Seminarios. Sociedad Espanola: Sociedad Espanola Mineralogia, p. 53-64.
- Bao, P., Li, G.-X. J. E. s., and technology, 2017, Sulfur-driven iron reduction coupled to anaerobic ammonium oxidation, v. 51, no. 12, p. 6691-6698.
- Beard, B. L., Johnson, C. M., Cox, L., Sun, H., Neilson, K. H., and Aguilar, C., 1999, Iron isotope biosignatures: Science, v. 285, no. 5435, p. 1889-1892.
- Bhatia, M. P., Kujawinski, E. B., Das, S. B., Breier, C. F., Henderson, P. B., and Charette, M. A., 2013, Greenland meltwater as a significant and potentially bioavailable source of iron to the ocean: Nature Geoscience, v. 6, no. 4, p. 274.
- Canfield, D. E., Thamdrup, B., and Hansen, J. W., 1993, The anaerobic degradation of organic matter in Danish coastal sediments: iron reduction, manganese reduction, and sulfate reduction: Geochimica et Cosmochimica Acta, v. 57, no. 16, p. 3867-3883.
- Crosby, H. A., Roden, E. E., Johnson, C. M., and Beard, B. L., 2007, The mechanisms of iron isotope fractionation produced during dissimilatory Fe (III) reduction by *Shewanella putrefaciens* and *Geobacter sulfurreducens*: Geobiology, v. 5, no. 2, p. 169-189.
- Dixon, J. B., Weed, S. B., and Parpitt, R., 1990, Minerals in soil environments: Soil Science, v. 150, no. 2, p. 562.
- Dong, H., Kostka, J. E., and Kim, J., 2003a, Microscopic evidence for microbial dissolution of smectite: Clays and Clay Minerals, v. 51, no. 5, p. 502-512.
- Dong, H., Kukkadapu, R. K., Fredrickson, J. K., Zachara, J. M., Kennedy, D. W., and Kostandarites, H. M., 2003b, Microbial reduction of structural Fe (III) in illite and goethite: Environmental Science & Technology, v. 37, no. 7, p. 1268-1276.
- Dos Santos Afonso, M., and Stumm, W., 1992, Reductive dissolution of iron (III)(hydr) oxides by hydrogen sulfide: Langmuir, v. 8, no. 6, p. 1671-1675.

- Eberl, D., and Velde, B., 1989, Beyond the Kübler index: Clay minerals, v. 24, no. 4, p. 571-577.
- Emerson, D., Scott, J. J., Benes, J., and Bowden, W. B., 2015, Microbial iron oxidation in the arctic tundra and its implications for biogeochemical cycling: *Appl. Environ. Microbiol.*, v. 81, no. 23, p. 8066-8075.
- Enning, D., and Garrelfs, J., 2014, Corrosion of iron by sulfate-reducing bacteria: new views of an old problem: *Appl. Environ. Microbiol.*, v. 80, no. 4, p. 1226-1236.
- Hawkings, J. R., Wadham, J. L., Tranter, M., Raiswell, R., Benning, L. G., Statham, P. J., Tedstone, A., Nienow, P., Lee, K., and Telling, J., 2014, Ice sheets as a significant source of highly reactive nanoparticulate iron to the oceans: *Nature communications*, v. 5.
- Icopini, G., Anbar, A., Ruebush, S., Tien, M., and Brantley, S., 2004, Iron isotope fractionation during microbial reduction of iron: the importance of adsorption: *Geology*, v. 32, no. 3, p. 205-208.
- Jaboyedoff, M., Bussy, F., Kübler, B., and Thelin, P., 2001, Illite "crystallinity" revisited: *Clays and clay minerals*, v. 49, no. 2, p. 156-167.
- Jaisi, D. P., Dong, H., and Liu, C., 2007, Influence of biogenic Fe (II) on the extent of microbial reduction of Fe (III) in clay minerals nontronite, illite, and chlorite: *Geochimica et Cosmochimica Acta*, v. 71, no. 5, p. 1145-1158.
- Jaisi, D. P., Kukkadapu, R. K., Eberl, D. D., and Dong, H., 2005, Control of Fe (III) site occupancy on the rate and extent of microbial reduction of Fe (III) in nontronite: *Geochimica et Cosmochimica Acta*, v. 69, no. 23, p. 5429-5440.
- Kim, K., Choi, W., Hoffmann, M. R., Yoon, H.-I., and Park, B.-K., 2010, Photoreductive dissolution of iron oxides trapped in ice and its environmental implications: *Environmental science & technology*, v. 44, no. 11, p. 4142-4148.
- Komadel, P., Madejová, J., and Stucki, J. W., 2006, Structural Fe (III) reduction in smectites: *Applied Clay Science*, v. 34, no. 1-4, p. 88-94.
- Kostka, J. E., Wu, J., Nealson, K. H., and Stucki, J. W., 1999, The impact of structural Fe (III) reduction by bacteria on the surface chemistry of smectite clay minerals: *Geochimica et Cosmochimica Acta*, v. 63, no. 22, p. 3705-3713.
- Li, Y.-L., Vali, H., Sears, S. K., Yang, J., Deng, B., and Zhang, C. L., 2004, Iron reduction and alteration of nontronite NAu-2 by a sulfate-reducing bacterium: *Geochimica et Cosmochimica Acta*, v. 68, no. 15, p. 3251-3260.
- Liu, C., Zachara, J. M., Foster, N. S., and Strickland, J., 2007, Kinetics of reductive dissolution of hematite by bio-reduced anthraquinone-2, 6-disulfonate: *Environmental science & technology*, v. 41, no. 22, p. 7730-7735.
- Monien, P., Lettmann, K. A., Monien, D., Asendorf, S., Wöfl, A.-C., Lim, C. H., Thal, J., Schnetger, B., and Brumsack, H.-J., 2014, Redox conditions and trace metal cycling in coastal sediments from the maritime Antarctic: *Geochimica et Cosmochimica Acta*, v. 141, p. 26-44.
- Petschick, R., Kuhn, G., and Gingele, F., 1996, Clay mineral distribution in surface sediments of the South Atlantic: sources, transport, and relation to oceanography: *Marine Geology*, v. 130, no. 3-4, p. 203-229.
- Raiswell, R., Benning, L. G., Tranter, M., and Tulaczyk, S., 2008, Bioavailable iron in the Southern Ocean: the significance of the iceberg conveyor belt: *Geochemical transactions*, v. 9, no. 1, p. 7.
- Reyes, C., Dellwig, O., Dähnke, K., Gehre, M., Noriega-Ortega, B. E., Böttcher, M. E., Meister, P., and Friedrich, M. W., 2016, Bacterial communities potentially involved in iron-cycling in Baltic Sea and North Sea sediments revealed by pyrosequencing: *FEMS microbiology ecology*, v. 92, no. 4, p. fiw054.
- Sainz-Díaz, C. I., Timón, V., Botella, V., Artacho, E., and Hernández-Laguna, A., 2002, Quantum mechanical calculations of dioctahedral 2: 1 phyllosilicates: Effect of

- octahedral cation distributions in pyrophyllite, illite, and smectite: *American Mineralogist*, v. 87, no. 7, p. 958-965.
- Stucki, J. W., and Kostka, J. E., 2006, Microbial reduction of iron in smectite: *Comptes Rendus Geoscience*, v. 338, no. 6, p. 468-475.
- Turner, D. R., and Hunter, K. A., 2001, *The biogeochemistry of iron in seawater*, Wiley Chichester, UK.
- Urrutia, M., Roden, E., Fredrickson, J., and Zachara, J., 1998, Microbial and surface chemistry controls on reduction of synthetic Fe (III) oxide minerals by the dissimilatory iron-reducing bacterium *Shewanella alga*: *Geomicrobiology Journal*, v. 15, no. 4, p. 269-291.
- Wei, N., and Finneran, K. T., 2011, Influence of ferric iron on complete dechlorination of trichloroethylene (TCE) to ethene: Fe (III) reduction does not always inhibit complete dechlorination: *Environmental science & technology*, v. 45, no. 17, p. 7422-7430.
- Zhang, J., Dong, H., Liu, D., Fischer, T. B., Wang, S., and Huang, L., 2012, Microbial reduction of Fe (III) in illite–smectite minerals by methanogen *Methanosarcina mazei*: *Chemical Geology*, v. 292, p. 35-44.

REVIEWERS' COMMENTS:

Reviewer #4 (Remarks to the Author):

The authors did a nice job in revising the manuscript. My concerns have been resolved.